# Visuomotor learning from postdictive motor error

**Jana Masselink\*, Markus Lappe**

Institute for Psychology and Otto Creutzfeldt Center for Cognitive and Behavioral Neuroscience, University of Muenster, Münster, Germany

**Abstract** Sensorimotor learning adapts motor output to maintain movement accuracy. For saccadic eye movements, learning also alters space perception, suggesting a dissociation between the performed saccade and its internal representation derived from corollary discharge (CD). This is critical since learning is commonly believed to be driven by CD-based visual prediction error. We estimate the internal saccade representation through pre- and trans-saccadic target localization, showing that it decouples from the actual saccade during learning. We present a model that explains motor and perceptual changes by collective plasticity of spatial target percept, motor command, and a forward dynamics model that transforms CD from motor into visuospatial coordinates. We show that learning does not follow visual prediction error but instead a postdictive update of space after saccade landing. We conclude that trans-saccadic space perception guides motor learning via CD-based postdiction of motor error under the assumption of a stable world.

## Introduction

Saccade motor control and visual space perception are inherently linked. We explore visual space via saccadic eye movements and, vice versa, adapt saccadic eye movements via visuospatial feedback. If a saccade falls repeatedly short of a target, the saccade vector gradually lengthens over several trials until the eyes land closer to the target again (*McLaughlin, 1967*; *Deubel et al., 1986*; *Wallman and Fuchs, 1998*; *Havermann and Lappe, 2010*; *Cassanello et al., 2019*). This plasticity guarantees saccade accuracy in the light of changing muscle dynamics, like short-term muscle fatigue, or physiological long-term changes during growth, aging, or disease (*Hopp and Fuchs, 2004*; *Pélisson et al., 2010*).

To understand the mechanism behind this remarkable capacity, a central question is which error signal drives learning. Early research focussed on the visual error, that is, the retinal distance of the post-saccadic target from the fovea. This appears to make sense because minimizing visual error would assure that the eye lands on target. However, saccades usually undershoot their target by 5–10% (*Robinson, 1973*; *Henson, 1979*; *Becker, 1989*) and do not fully compensate for intra-saccadic target shifts (*Deubel et al., 1986*; *Straube et al., 1997*; *Wallman and Fuchs, 1998*; *Noto et al., 1999*), the technique often used in studies of saccadic adaptation. This suggests that the oculomotor system tolerates some systematic visual error. Moreover, recent studies showed that saccades become shorter when the target is artificially shifted closer to the fovea during the saccade — thus increasing visual error during learning (*Wong and Shelhamer, 2011*) — and that saccadic adaptation occurs even for microscopic saccades in which the target never leaves the fovea and the visual error is always zero (*Havermann et al., 2014*).

More recent studies proposed that, instead of minimizing visual error, learning minimizes visual prediction error (*Bahcall and Kowler, 2000*; *Wong and Shelhamer, 2011*; *Collins and Wallman, 2012*). Visual prediction error is the deviation of the post-saccadic visual error, that is, the post-saccadic retinal target position, from an internally predicted visual error, that is, an error that the oculomotor system expects to occur.

\*For correspondence:
jana.masselink@uni-muenster.de

**Competing interests:** The authors declare that no competing interests exist.

For saccadic eye movements, learning from post-saccadic spatial feedback is critical since visual input is suppressed during the movement such that saccades can neither be tracked nor corrected by visual feedback online (*Volkmann et al., 1968*; *Bridgeman et al., 1975*; *Li and Matin, 1990*). Moreover, saccades are so brief that latencies in the visual system postpone any visual input to after the saccade. Hence, the predicted visual error is supposed to rely on an update of the pre-saccadic target position by internal information about the upcoming saccade. This information is derived from a copy of the motor command, known as corollary discharge (*Duhamel et al., 1992*; *Umeno and Goldberg, 1997*; *Crapse and Sommer, 2008b*; *Melcher and Colby, 2008*; *Crapse and Sommer, 2009*). It informs visual areas about the eye movement in order to support our percept of a stable external world (*von Helmholtz, 1867*; *Sperry, 1950*; *von Holst and Mittelstaedt, 1950*; *Bridgeman and Stark, 1991*; *Wurtz, 2008*; *Zimmermann et al., 2018*). The corollary discharge is assumed to be processed via several pathways. The most prominent pathway extends from superior colliculus (SC) via MD thalamus to the frontal eye fields (FEF; *Sommer and Wurtz, 2002*; *Sommer and Wurtz, 2004a*; *Sommer and Wurtz, 2004b*; *Sommer and Wurtz, 2006*; *Cavanaugh et al., 2020*). Other pathways extend from SC via the thalamic pulvinar to parietal and occipital cortex (*Wurtz et al., 2011*; *Berman et al., 2017*), from the cerebellum via the ventrolateral thalamus to frontal cortex (*Middleton and Strick, 2000*; *Gaymard et al., 2001*; *Zimmermann et al., 2015*) and back from FEF through the basal ganglia to SC (*Sommer and Wurtz, 2008*; *Wurtz, 2008*).

Along these pathways, the information provided by the corollary discharge needs to be transformed by a forward dynamics model from motor to visual coordinates before it can be used by vision (*Bays and Wolpert, 2007*; *Sommer and Wurtz, 2008*; *Crapse and Sommer, 2008b*; *Franklin and Wolpert, 2011*). To accurately estimate the visual effect of the saccade, that is, to compute the displacement of visual space due to the saccade, this transformation should rely on the current dynamics of the eye muscles, for example whether the eye muscles are fatigued or strong (*Bays and Wolpert, 2007*; *Shadmehr et al., 2010*; *Franklin and Wolpert, 2011*). We thus need to distinguish between the corollary discharge signal in motor coordinates, abbreviated as $CD_M$, which provides the input into the forward dynamics model, and the computed displacement of visual space, abbreviated as $CD_V$, which is the output of the forward dynamics model and describes the expected effect of the saccade on visual coordinates. $CD_V$ is believed important for trans-saccadic visual localization, the estimate of the post-saccadic position of a target that was seen before the saccade (*Bahcall and Kowler, 1999*; *Sommer and Wurtz, 2008*; *Wurtz, 2008*; *Cavanaugh et al., 2016*; *Wurtz, 2018*; *Binda and Morrone, 2018*). If the forward dynamics model does not adequately transform $CD_M$ into $CD_V$, that is, if the transformation has non-unity gain, errors in trans-saccadic localization appear.

Studies have indicated that not only the saccade vector but also visual localization changes during learning. This occurs (a) during fixation (*Moidell and Bedell, 1988*; *Collins et al., 2007*; *Hernandez et al., 2008*; *Schnier et al., 2010*; *Zimmerman and Lappe, 2010*; *Gremmler et al., 2014*) and (b) even stronger after adapted saccades (*Bahcall and Kowler, 1999*; *Collins et al., 2007*; *Zimmermann and Lappe, 2009*; *Schnier et al., 2010*; *Klingenhoefer and Bremmer, 2011*). First, (a) suggests that changes in the saccade vector combine adaptation of visual target position and adaptation of the visual-to-motor transformation, that is, the inverse model that derives the motor command. Second, the discrepancy in target localization between (a) and (b) suggests that $CD_V$, that is, the computed displacement of visual space due to the saccade — estimated by the forward dynamics model — might become biased during learning. If $CD_V$ were accurate, the post-saccadic target should be predicted exactly where it was perceived before the saccade in external space, and no bias should occur. The bias in trans-saccadic target perception is known from other studies apart from saccadic learning, namely when some stage of the CD pathway is lesioned (*Ostendorf et al., 2010*) or experimentally perturbed (*White and Snyder, 2007*; *Prime et al., 2010*; *Ostendorf et al., 2012*; *Cavanaugh et al., 2016*). For example, *Cavanaugh et al., 2016* showed that inactivation of MD thalamus in the macaque monkey causes a shift in target localization after saccade landing – consistent with a deficient $CD_V$ and an erroneous prediction of post-saccadic retinal target location. Third, (b) questions whether actual and predicted visual error match in the adapted steady state, that is, when learning reaches saturation. If learning were driven by visual prediction error, actual and predicted visual error were expected to match.

In the present study, we hypothesize that saccadic motor learning relies on multiple plasticity within the visuomotor circuitry – comprising adaptation of visual target position, of the inverse model (i.e. the visual-to-motor transformation to derive the motor command), and of the forward dynamics model (i.e. the motor-to-visual transformation to derive the the computed displacement of visual space, $CD_V$, resulting form the eye movement). We estimate the state of $CD_V$ during learning by comparing pre-and post-saccadic target localization. We then examine which learning rule explains this plasticity – comparing a model that minimizes visual prediction error with a model following a novel learning approach which we term postdictive motor error learning. According to this framework, the visuomotor system learns from a postdictive update of pre-saccadic target position based on $CD_V$. We show, first, that visual target position and forward dynamics model (i.e. $CD_V$) collectively learn from error together with the inverse model (i.e. the motor command) and, second, that this error relies on a postdictive update of space after movement completion. Initially, $CD_V$ was hypometric, consistent with saccade hypometry. During learning, $CD_V$ dissociated from the saccade, consistent with incomplete motor compensation. Our results reveal that learning occurs under the explicit assumption of a stable world and not in response to its violation.

## Experimental methods

In order to provide a broad database for the modeling we measured four different learning conditions that are known to produce different amounts of change in the saccade vector and in visual localization. Each learning condition consisted of 280 saccade trials requiring a reactive saccade to a 13° rightward target (*Figure 1*). In two conditions with constant target step (abbreviated as CTS conditions; *McLaughlin, 1967*), the target was shifted either 3° inward (opposite to saccade direction, CTS$_{in}$) or outward (in saccade direction, CTS$_{out}$) during saccade execution. In two conditions with constant visual error (abbreviated as CVE conditions; *Robinson et al., 2003*; *Havermann and Lappe, 2010*; *Zimmerman and Lappe, 2010*), the target was shifted to the position that is 3° inward (CVE$_{in}$) or outward (CVE$_{out}$) of the post-saccadic gaze direction. Every 70 saccade trials, we quantified the state of the visuomotor system with a probe block measuring the saccade vector, the localization of a target during fixation (referred to as pre-saccadic localization) and the localization of a pre-saccadically presented target after saccade landing (referred to as post-saccadic localization; probe block 1 before learning, probe blocks 2–4 during learning, probe block 5 after learning).

Learning direction (inward, outward) and paradigm (CTS, CVE) were varied to ensure the generalizability of our modeling results. Our model should be applicable to both learning directions, for example capturing more learning in the saccade vector to be expected for inward compared to outward target steps (*Kojima et al., 2004*; *Panouillères et al., 2009*; *Zimmerman and Lappe, 2010*; *Pélisson et al., 2010*). Moreover, besides the classical CTS paradigm, our model should explain why changes in saccade vector and target localization converge even if the visual error cannot be reduced in the CVE paradigm (*Robinson et al., 2003*; *Havermann and Lappe, 2010*; *Zimmerman and Lappe, 2010*). In either paradigm, the target step manipulates the visual position of the post-saccadic target (visual error) that is used in preparing the error signal that supports learning. Learning in our model is driven by error reduction and aims for steady states with zero error. This is the aim in both paradigms – whether the visual error can be reduced by learning in the CTS conditions or whether the visual error cannot be reduced by learning in the CVE conditions. A target step of 3° (23% of the pre-saccadic target distance in the CTS paradigm) was chosen to ensure a sufficient amount of learning in the saccade vector (*McLaughlin, 1967*; *Deubel et al., 1986*; *Umeno and Goldberg, 1997*; *Havermann and Lappe, 2010*).

### Subjects

Data were recorded from two samples of each *N* = 18 healthy subjects (36 subjects in total). All subjects had normal or corrected-to-normal vision and were naïve to the objectives of the experiment. Sample 1 performed the inward conditions (CTS$_{in}$ and CVE$_{in}$, 21.2 ± 2.8 years, two male) and sample 2 performed the outward conditions (CTS$_{out}$ and CVE$_{out}$, 24.7 ± 6.7 years, four male). One subject was excluded from sample 1 because pre- and post-saccadic localizations consistently deviated more than five standard deviations from the mean over subjects. All subjects gave written informed consent prior to participation. The experiment was approved by the ethics committee of the Department of Psychology and Sport Science of the University of Münster.

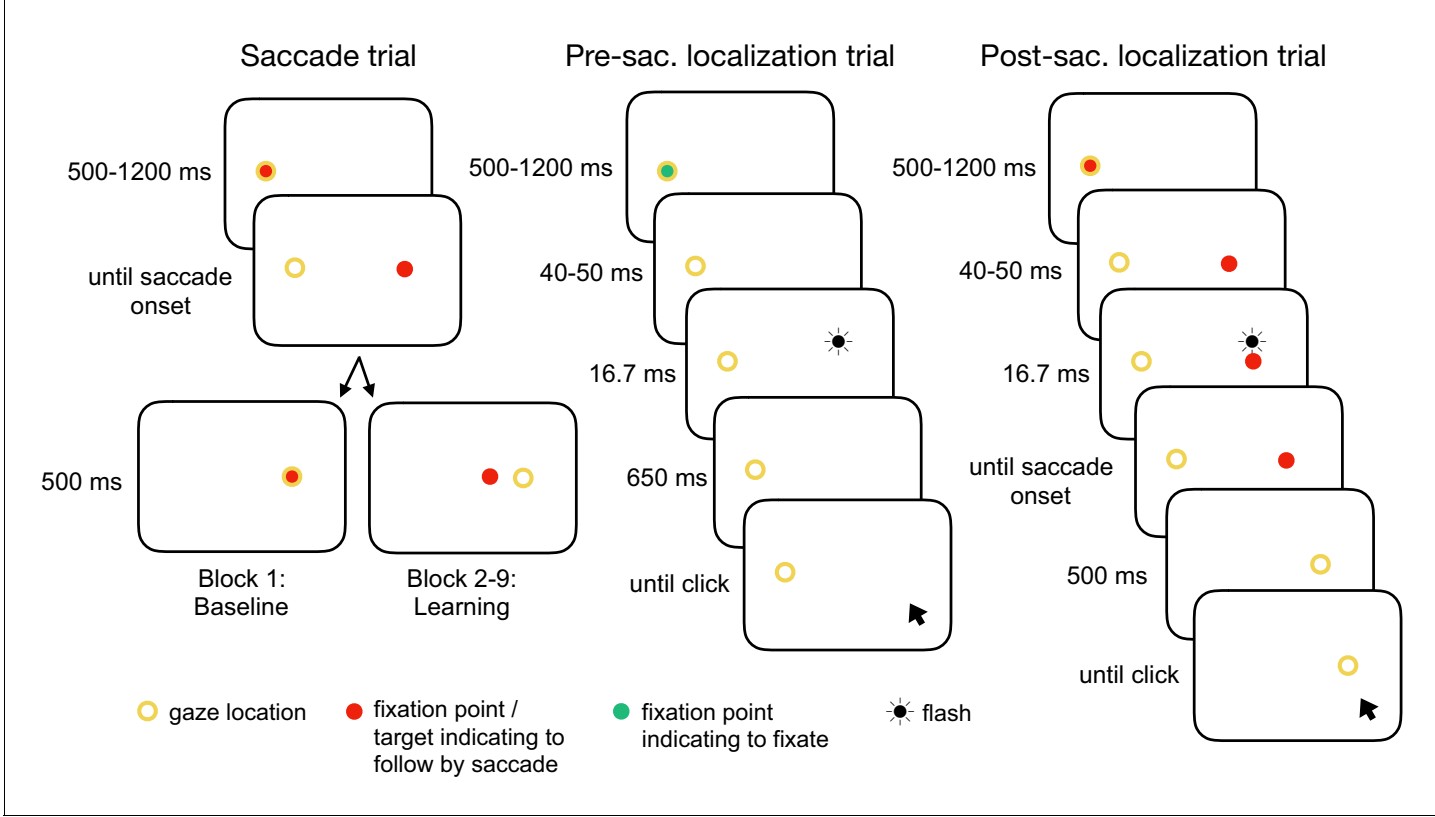

**Figure 1.** Experimental tasks. In the saccade trials, subjects performed a reactive saccade to a 13° rightward target. During saccade execution from block 2 onwards, the target was shifted to a new location depending on the learning condition, e.g. 3° inward as shown here (CTS$_{in}$). In the pre-saccadic localization trials, subjects hold their gaze at the fixation point while localizing a white 16.7 ms flash with a gray dot cursor. In the post-saccadic localization trials, subjects performed a saccade as in the saccade trials but hold their gaze after saccade landing and report the location of the pre-saccadic flash that had appeared 40–50 ms after target onset. The yellow circle illustrates gaze location but was not present at the stimulus monitor.

## Setup

Subjects were seated with a distance of 62 cm in front of an Eizo FlexScan F930 monitor (Eizo, Hakusan, Japan; 800 × 600 pixels, 120 Hz) with a visual field of 32.8° × 25.8° (40 cm ×30 cm). Their head was restrained with a chin rest, forehead support and a head belt. The room was completely dark with all sources of light eliminated to avoid the use of visual references in the localization tasks (room luminance below 0.01 cd/m²). The monitor was covered with a dark gray foil that reduced luminance by two log units. This was done to prevent visibility of monitor background light and contrast to the surroundings (remaining background luminance below 0.01 cd/m²) and to reduce effects of phosphor persistence (*Georg and Lappe, 2009*; *Zimmermann and Lappe, 2009*; *Zimmerman and Lappe, 2010*; *Schnier et al., 2010*). To report perceived locations, subjects operated a mouse cursor via a multi-touch trackpad (Apple Inc, Cupertino, CA) with their right index finger.

The position of the right eye was recorded at 1000 Hz using an Eyelink 1000 (SR Research, Ontario, Canada). Calibration was performed with a white nine-point grid on black background. For online detection of saccade onset, position threshold was set to 2.5° and velocity threshold to 22 $\frac{°}{s}$. We chose this rather liberal velocity threshold in order to perform peri-saccadic stimulus changes as early as possible to avoid any afterglow at saccade landing. The display change occurred 23.3 ± 1.5 ms after the offline detected saccade onset and 31.3 ± 6.3 ms before the offline detected saccade offset (mean and SD across subject means). This timepoint was determined from the Eyelink EDF data files that received an event message command from the Matlab script after the display change was performed. Saccade landing was detected online as soon as saccade velocity fell below 30 $\frac{°}{s}$. This threshold allowed the best temporal and spatial accuracy of the post-saccadic target with

respect to saccade landing time and position in the CVE conditions. The experimental procedure was controlled by a Matlab script (Mathworks, Natick, MA) using the Psychophysics Toolbox.

## Design

Each condition contained three trial types arranged in nine blocks (*Figure 1*). The even block numbers 2, 4, 6, and 8 induced visuomotor learning (saccade trials) while the odd block numbers 1, 3, 5, 7, and 9 probed the current state of saccade vector, pre-and post-saccadic localization. The probe blocks were interspersed with saccade trials to preserve the current learning state (similar to *Bahcall and Kowler, 1999*, no target step in block 1). Each trial began with a fixation point presented 6.5° left of the screen center, measuring 0.5° in diameter. Its color indicated whether subjects needed to perform a saccade (red, saccade trials and post-saccadic localization trials) or to keep fixation on the fixation point (green, pre-saccadic localization trials). The trial was initiated by disappearance of the fixation point if the subject had fixated it for a randomly selected time interval drawn from a uniform distribution between 500 and 1200 ms, using a position threshold of 2.5°. This liberal threshold was chosen because in complete darkness, the detection of vertical gaze location can be very sensitive to small changes in pupil size as light incidence varies with stimulus presentation. Please note that drifts of gaze position due to changes in pupil size appear mainly in vertical direction and less in horizontal direction (*Drewes et al., 2014*; *Choe et al., 2016*).

### Saccade trials

Simultaneous with fixation point offset, a red target of 0.5° diameter appeared 13° to the right of the fixation point. Subjects were instructed to look at it as fast and as accurately as possible. In the CTS conditions, the target was stepped 3° inward ($CTS_{in}$) or outward ($CTS_{out}$) of the initial target position as soon as saccade onset was detected. In the CVE conditions, the target was deleted with saccade onset (to avoid afterglow at saccade landing) and reappeared 3° inward ($CVE_{in}$) or outward ($CVE_{out}$) of the saccade landing position as soon as saccade landing was detected. The target was shown for 500 ms after saccade landing. As an exception, the target stayed at its initial position in probe block 1 (baseline). These trials were aimed to prevent the typical saccade vector decline in the absence of a post-saccadic target. Hence, probe block 1 was equal between all learning conditions.

### Post-saccadic localization trials

The post-saccadic localization trials started with the same target as the saccade trials. Then, 40–50 ms after target onset, a white dot was flashed for 16.7 ms (two monitor refreshes, 2° above the target, 0.5° in diameter). The flash was presented with the same constant horizontal eccentricity as the saccade target such that its horizontal localization judgement could be matched to the visual target localization. Please note that any variation of the flash position from the target position on the horizontal axis would have implied localization transfer to a different horizontal eccentricity and, hence, would likely have diminished the effects in the pre-saccadic localization which are usually rather small and difficult to measure (*Moidell and Bedell, 1988*; *Collins et al., 2007*; *Hernandez et al., 2008*; *Schnier et al., 2010*; *Zimmerman and Lappe, 2010*; *Gremmler et al., 2014*). The target was extinguished as soon as the saccade onset was detected. Subjects were instructed to aim their gaze at the target as fast and as accurately as possible and to stay fixated in the dark at the saccade landing location. If gaze deviated from the saccade landing location more than 4°, a beep tone was presented until gaze position returned to the accepted fixation area. On average, a beep tone occurred in 23.4 ± 15.3% of trials, consistent with the fixation in darkness being a demanding task. A gray dot cursor appeared 500 ms after saccade landing (0.7° in diameter at a random position drawn from a uniform distribution between 15.9° and 20.9° rightward from the fixation point and 4° above the lower monitor border). Subjects clicked the cursor at the perceived flash position while still fixating at the saccade landing location. On average, subjects started the saccade with a latency of 218.9 ± 23.8 ms and a duration of 55.7 ± 5.3 ms and clicked at the perceived position 2320 ± 571 ms after saccade landing (mean and SD over subjects). In case they did not perceive the flash, they were asked to click at the lowest position possible (the invisible lower screen border).

### Pre-saccadic localization trials

A green fixation point indicated that subjects needed to stay fixated at the fixation point location in the dark even after fixation point offset. If gaze deviated from the fixation point location more than 4°, a beep tone was presented until gaze position returned to the accepted fixation area. On average, this was the case in $20.7 \pm 12.3\%$ of trials. Analogous to the post-saccadic localization trials, a white dot was flashed 40–50 ms after fixation point offset. A gray dot cursor appeared 710 ms after fixation point offset. Subjects had to click the cursor at the perceived flash position (or at the lowest position possible if they had not perceived the flash) while remaining fixated at the invisible fixation point location. On average, they performed the cursor click $2460 \pm 420$ ms after flash offset. Flash and dot cursor parameters were the same as in the post-saccadic localization trials. As the fixation point turned off before flash onset and the cursor appeared after flash offset, there were no visual references that could affect the perceived flash position.

At the start of each session, subjects practiced every trial type (saccade trials without target step) until they felt confident with the task and successfully hold their gaze in the dark in the localization trials. Each learning block consisted of 70 saccade trials, resulting in 4*70 = 280 trials in total (refresh saccade trials of the probe blocks excluded). The probe blocks contained a repeated sequence of a pre-saccadic localization trial, a refresh saccade trial, a post-saccadic localization trial and another refresh saccade trial. In the first and the last probe block (block 1 and 9), the sequence was repeated 21 times minus the last refresh saccade trial, resulting in 4*21–1 = 83 trials, containing 21 pre- and 21 post-saccadic localization trials. To avoid a long interruption of the ongoing learning process, the other probe blocks (blocks 3, 5, and 7) consisted of nine sequence repetitions minus the last refresh saccade trial, resulting in 4*9–1 = 35 trials, containing nine pre- and nine post-saccadic localization trials. In sum, each session comprised 4*70 + 2*83 + 3*35 = 551 trials with an inter-trial interval of 800 ms within the blocks. Sessions took around 45 min each and were counterbalanced across participants. The two different sessions for each participant were recorded at least 14 days apart to prevent carryover effects from the first to the second session. Testing for carryover effects via t-tests (first vs. second session) separately for baseline saccade amplitudes, pre- and post-saccadic localizations did not reveal any significant effects.

## Data processing

Data analysis was performed offline in Matlab R2017a (Mathworks, Natick, MA). From the post-saccadic localization and saccade trials, we selected the rightward primary saccades with a latency between 100 and 400 ms (reactive saccades) and a horizontal saccade vector of at least 5°. Saccade start and end point were detected by a combined velocity and acceleration criterion and were visualized for inspection together with the position trace. The saccade vectors of the saccade trials were plotted to monitor the course of saccade vector learning but only the saccade vectors of the post-saccadic localization trials were used for further analysis.

Pre- and post-saccadic localizations were quantified as the distance of the cursor click from the fixation point. Post-saccadic localizations were accepted in case of a valid primary saccade that did not start earlier than 100 ms after flash offset. This was done to avoid localization errors due to peri-saccadic compression (*Ross et al., 1997*; *Lappe et al., 2000*). Moreover, post-saccadic localizations were only accepted if, after the saccade, gaze was successfully held at the position of saccade landing. Pre-saccadic localizations were accepted if no saccade was performed within the first 400 ms after flash onset. Pre- and post-saccadic localizations of trials in which subjects had clicked within the lower 30% of the display or deviated from fixation (at the fixation point in pre-saccadic localizations and at the saccadic landing point in post-saccadic localizations) for more than 1400 ms were omitted from analysis. Based on these criteria, $89.2 \pm 5.3\%$ of the pre-saccadic localization trials and $76.7 \pm 14.3\%$ of the post-saccadic localization trials (saccade vector and post-saccadic localizations) were used for further analysis.

For each of the five probe blocks within a session, we calculated the median saccade vector, median pre- and median post-saccadic localization, excluding outliers with more than three scaled median absolute deviations from the median. In contrast to the saccade vectors within the learning blocks, we did not observe any systematic changes of saccade vectors, pre- or post-saccadic localizations within the probe blocks. Hence, for model fitting, we considered the median of each probe

block to reflect the current state of the system at trials $n = [1, 71, 141, 211, 281]$ of 281 pure saccade trials in total.

## Model

The modeling was aimed to determine, first, the degree to which plasticity of the visual gain, the motor gain (inverse model) and the CD gain (forward dynamics model) contributes to the maintenance of visuomotor consistency and oculomotor control and, second, the nature of the error signal that is used. Therefore, we set up two state-space models that describe the saccadic circuitry with the same basic visuomotor transformations but differ in the error signal that drives learning. However, in order to formalize multiple plasticity for the specific motor system of saccades, we first explain our general model conception as a basis for the subsequent model equations for the respective error signals before we provide details on the implementation.

### Model conception

The basis of our modeling approach is that visuospatial and motor signals are represented in different coordinate systems which are interconnected via sensorimotor transformations from one coordinate system to the other (*Figure 2*). Learning relies on synaptic plasticity and is hence supposed to be driven by changes in sensorimotor transformations (synaptic gains; *Shadmehr et al., 2010*; *Wolpert et al., 2011*). As spatial target errors indicate a mismatch between motor behavior and perceived space, learning for error reduction relies on recalibration of these sensorimotor transformations (*Hopp and Fuchs, 2004*; *Bays and Wolpert, 2007*; *Franklin and Wolpert, 2011*). This recalibration aims to minimize error and should ideally proceed until the error is nullified, in which case the sensorimotor transformations correctly represent the environment (spatial target positions) and the internal state of the motor system (the current eye dynamics). We are interested in determining the potential contribution of plasticity at all stages of the transformation. For simplicity we will model these changes simply as changes of gain, that is, as single scalar values by which the eccentricity or amplitude of the respective signal is scaled. Three sensorimotor gains are allowed to learn from error in the model - a visual gain $\omega_v$ to transform retinal input into target position on the spatial map, a motor gain $\omega_m$ to transform spatial target position into a motor command (inverse model), and a CD gain $\omega_{cd}$ to transform the corollary discharge $CD_M$ of the motor command into $CD_V$, the computed displacement of visual space due to the saccade (forward dynamics model). We describe the motivation for and use of each gain in the following.

#### Modeling visual and motor gains

Plasticity of the visual gain $\omega_v$ occurs as perception of the pre-saccadic target localization changes during learning (*Moidell and Bedell, 1988*; *Collins et al., 2007*; *Hernandez et al., 2008*; *Schnier et al., 2010*; *Zimmerman and Lappe, 2010*; *Gremmler et al., 2014*). Hence, it reflects a recalibration of the mapping of the retinal target input onto the perceived spatial target location when errors are assigned to an internal failure of spatial target representation (*Collins et al., 2007*; *Zimmerman and Lappe, 2010*; *Gremmler et al., 2014*).

Plasticity of the motor gain $\omega_m$ occurs if the saccade vector requires a different motor command, depending on the current dynamics of the eye muscles. In the brain, the visual-to-motor transformation from spatial target percept to motor command is performed by a dynamic inverse model that needs to be highly plastic (*Bays and Wolpert, 2007*; *Franklin and Wolpert, 2011*). For example, if the eye muscles fatigue such that the saccade falls short, the inverse model increases the motor gain $\omega_m$ to increase the saccade vector and better aim for the pre-saccadic target localization. We will use the pre-saccadic target localization as the definition of the saccade goal. While spatial target localization and saccade landing roughly match in baseline saccades (*Müsseler et al., 1999*; *Lappe et al., 2000*; *Stork et al., 2010*), they have been found to diverge during learning from peri-saccadic target steps (*Moidell and Bedell, 1988*; *Collins et al., 2007*; *Hernandez et al., 2008*; *Schnier et al., 2010*; *Zimmerman and Lappe, 2010*; *Gremmler et al., 2014*).

During learning, the spatial target code remains roughly constant in the lateral intraparietal area (LIP; *Goldberg et al., 2002*; *Steenrod et al., 2013*) and the motor map of the superior colliculus (SC; *Frens and Van Opstal, 1997*; *Edelman and Goldberg, 2002*; *Quessy et al., 2010*). Therefore, changes in saccade vector with respect to spatial target percept likely reflect downstream changes

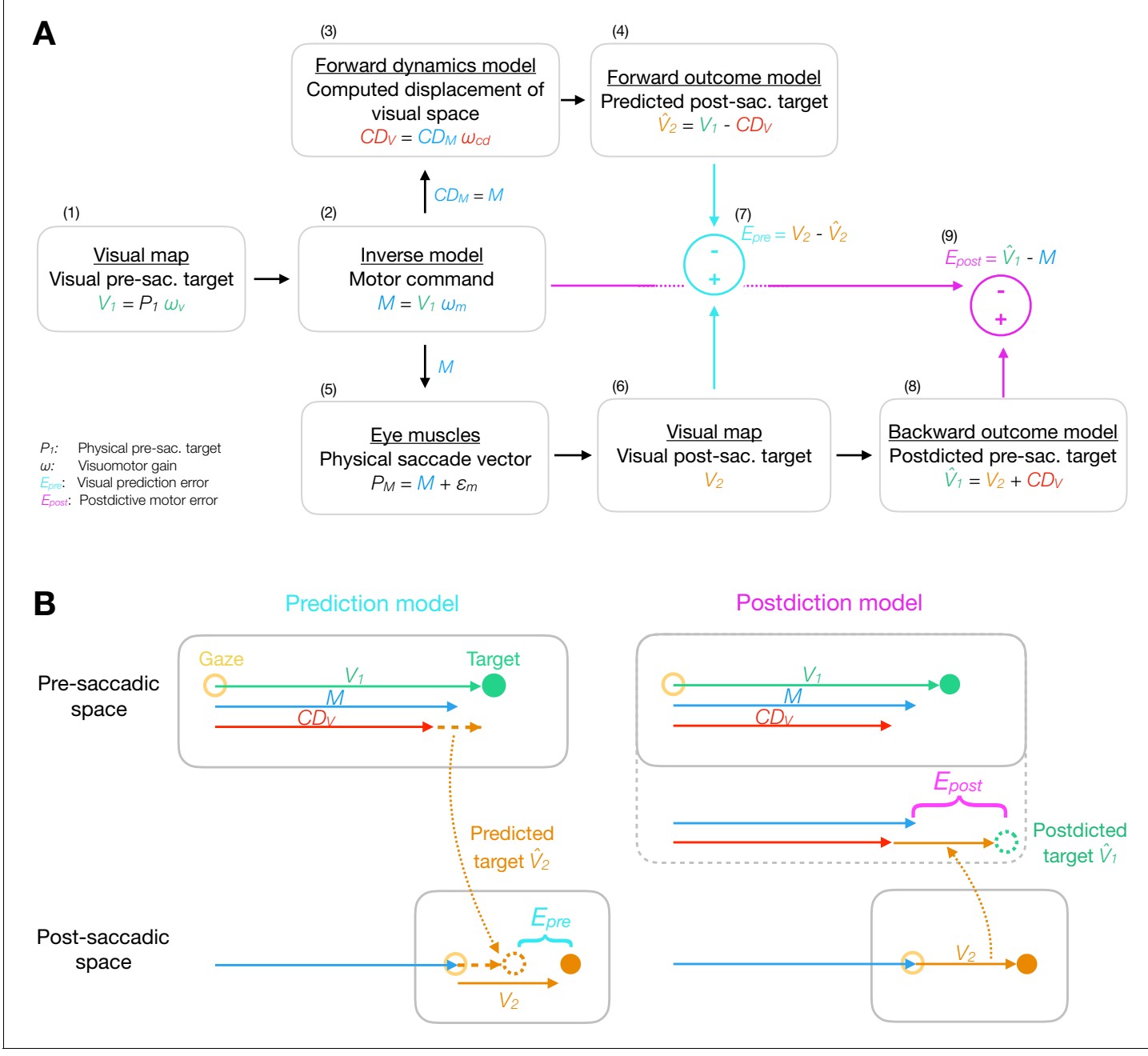

**Figure 2.** Model framework. (**A**) The visual pre-saccadic target $V_1$ (i.e. the perceived location of a physical target $P_1$) is transformed into the motor command $M$. Before saccade execution, a forward dynamics model transforms a copy of the motor command $CD_M$ into a computed displacement of visual space $CD_V$, a visual estimate of the saccade vector. Hence, the forward outcome model predicts the visual post-saccadic target to appear at position $\hat{V}_2$. After saccade execution, the visual post-saccadic target appears at position $V_2$. According to prediction-based learning, the visuomotor system detects an error if the visual post-saccadic target deviates from its prediction ($E_{pre}$), experiencing a violation of spatial stability. According to postdiction-based learning, the visuomotor system assumes the world to remain stable during the saccade. Hence, a backward outcome model postdicts the visual post-saccadic target back to pre-saccadic space ($\hat{V}_1$) in order to retroactively evaluate the motor command ($E_{post}$). (**B**) Computation of visual prediction error $E_{pre}$ and postdictive motor error $E_{post}$. In prediction-based learning, the predicted visual error $\hat{V}_2$ is derived from the visual pre-saccadic target $V_1$ and the $CD_V$ signal in pre-saccadic space ($\hat{V}_2 = V_1 - CD_V$) and is then compared to the actual visual error in post-saccadic space ($E_{pre} = V_2 - \hat{V}_2$). In postdiction-based learning, the visual error $V_2$ is obtained in post-saccadic space and postdicted back to pre-saccadic space based on the $CD_V$ signal ($\hat{V}_1 = V_2 + CD_V$). It is then compared to the original motor command to calculate postdictive motor error ($E_{post} = \hat{V}_1 - M$).

of the motor command, for example in the cerebellum (*Bays and Wolpert, 2007*; *Franklin and Wolpert, 2011*; *Taylor and Ivry, 2014*). This means that the saccade error is assigned to internal motor failure, that is, to a change in eye dynamics reflected by the motor gain $\omega_m$. Such credit assignment to muscle dynamics is essential for motor learning in saccades – during fatigue in natural saccades or with an unperceived peri-saccadic target step in the laboratory (*Volkmann et al., 1968*; *Bridgeman et al., 1975*; *Li and Matin, 1990*). This is a crucial difference between saccade learning and other motor learning tasks. For example, in manual motor learning target perturbations do not remain undetected (*Shadmehr et al., 2010*; *Michel et al., 2018*) and the spatial movement goal and the movement vector may change synchronously.

## Modeling CD gain

The need for plasticity of the CD gain $\omega_{cd}$ in the forward dynamics model directly results from the plasticity of the motor gain $\omega_m$. The CD gain describes the transformation in the forward dynamics model of $CD_M$, the copy of the motor command, into $CD_V$, the computed displacement of visual space. This transformation depends on an internal model of the current eye dynamics as it aims to predict how much visual space is going to change when the current saccade is performed. The transformation is necessary before corollary discharge information can be used by the visual system (*Bays and Wolpert, 2007*; *Sommer and Wurtz, 2008a*; *Crapse and Sommer, 2008a*; *Franklin and Wolpert, 2011*). For example, in response to muscle fatigue that produces an outward error, the motor gain of the inverse model which provides the visual-to-motor transformation increases in order to keep the saccade on target. To compensate for the increased motor gain, the CD gain of the forward dynamics model must decrease in order not to overestimate the saccade vector from the now stronger motor command (*Crapse and Sommer, 2008a*; *Thier and Markanday, 2019*). Indeed, CD gain can be plastic and deviate from the actual oculomotor behavior (*Haarmeier and Thier, 1996*; *Haarmeier et al., 1997*). If the decrease of the CD gain does not exactly mirror the increase of the motor gain, $CD_V$ deviates from the outcome of the motor command.

For our model, we quantify the CD gain from the difference between pre- and post-saccadic localization with respect to the saccade vector. For example, if the CD gain underestimates the saccade vector, the post-saccadic localization will be further outward than the pre-saccadic localization. If the CD gain correctly reflects the saccade vector, the target will be reported at the same location before and after the saccade in external space. This is nearly the case in baseline saccades, pointing toward a roughly intact CD gain under natural viewing conditions (*Collins et al., 2009*; *Zimmerman and Lappe, 2010*; *Collins, 2014*). It has been argued that during learning the CD signal continues to reflect the baseline saccade, although the actual saccade adapts, such that post-saccadic target localization appears shifted in learning direction (*Bahcall and Kowler, 1999*; *Collins et al., 2009*). However, the $CD_V$ signal does not need to either stay at baseline level or correctly reflect the actual saccade. It can as well reflect an intermediate state, which we quantify in our model based on the trans-saccadic target localization data and which we describe by the CD gain.

## Modeling plasticity as a continuous error minimization process in visuomotor function

In order to serve an accurate calibration of motor performance with spatial perception, we assume that the error is calculated after every single saccade and adaptive processes are continuously active to reduce error (*Srimal et al., 2008*; *Herzfeld et al., 2018*; *Cassanello et al., 2019*; *Wolpert et al., 2011*). This implies that a steady state is reached when the error is nullified. We assume that this is the case in baseline saccades and after learning has fully converged. Hence, we aim to explain calibration of baseline visuomotor behavior and learning by a unified account that allows a flexible transition of the visuomotor system to a new steady state if errors occur, for example in response to a peri-saccadic target step. Current models often describe motor learning as a deviation from a preset baseline state, including a trial-by-trial decay rate that pulls behavior back to the baseline level (*Chen-Harris et al., 2008*; *Xu-Wilson et al., 2009*; *Albert and Shadmehr, 2018*). In this case, a steady state is reached if learning and decay are in balance but error minimization remains incomplete. Such models need to specify how and with what gains the baseline level is determined. In the special case of saccades, where the target step in the double-step experiments is not perceived (*Volkmann et al., 1968*; *Bridgeman et al., 1975*; *Li and Matin, 1990*), it is unclear how learning of

motor and perceptual gains should be restrained by a pre-defined baseline state. It seems implausible that the visuomotor system encompasses an innate status quo of sensorimotor transformations. Instead, for our model, we assume that learning serves optimal motor behavior and proceeds until error nullification. Thus, we explicitly search for an error signal that provides a universal explanation for visuomotor steady states without the need for a pre-defined status quo. Hence, this error serves to continuously optimize the alignment of motor performance with visual perception (*Zimmerman and Lappe, 2010*; *Havermann et al., 2014*; *Zimmermann and Lappe, 2016*).

## Modeling different types of error

In our modeling, we compare two error signals that relate the post-saccadic visual error to the pre-saccadic scene based on the $CD_V$ signal. One of them is the visual prediction error (*Bahcall and Kowler, 2000*; *Wong and Shelhamer, 2011*; *Collins and Wallman, 2012*), that is, the difference between the predicted and the actual location of the target on the retina after the saccade. This error aims to minimize discrepancies between outcome and prediction, rather than aiming to bring the eye on target. The other is a novel proposal which we term the postdictive motor error. Postdiction describes a backward modeling process that transforms the post-saccadic visual target into pre-saccadic coordinates using the $CD_V$ signal. Hence, postdiction updates the internal representation about where the pre-saccadic target actually appeared. The postdictive motor error compares this position to the performed motor command, aiming to bring the eye close to the target while also keeping vision, motor control, and CD processing in register with each other. *Figure 2* presents the rationale of the basic model as well as the computation of visual prediction error $E_{pre}$ (prediction model) and postdictive motor error $E_{post}$ (postdiction model).

One may also consider the pure post-saccadic visual error, that is, the distance of the post-saccadic target from the fovea after the saccade. However, several critical observations in the literature show that the visual error alone cannot drive learning. First, learning is driven by error reduction and aims for steady states with error nullification. However, it was repeatedly shown that the visuomotor system accepts a remaining amount of visual error at steady state (saccades tend to undershoot their target by 5-10%, *Robinson, 1973*; *Henson, 1979*; *Becker, 1989*) and after learning from peri-saccadic target shifts (saccade gain adaptation does usually not fully reach the target; *Deubel et al., 1986*; *Straube et al., 1997*; *Wallman and Fuchs, 1998*; *Noto et al., 1999*). Second, minimization of visual error would predict the visuomotor system to learn until the saccade lands on the post-saccadic target in the CTS paradigm. However, learning converges at an earlier stage (*Moidell and Bedell, 1988*; *Deubel et al., 1986*; *Straube et al., 1997*; *Wallman and Fuchs, 1998*; *Schnier et al., 2010*). Third, minimization of visual error would predict endless learning in the CVE paradigm, in which the target is shifted with a constant distance to the post-saccadic gaze location, thus keeping the visual error constant. Instead, learning converges at some stage even in this paradigm (*Robinson et al., 2003*; *Zimmerman and Lappe, 2010*; *Havermann and Lappe, 2010*). Fourth, if saccades purposely undershoot their target by means of accepting a certain amount of visual error, this undershoot should be actively maintained during learning. However, learning from inward stepping targets converges with a remaining visual error in opposing direction of the primary saccade (*Kojima et al., 2004*; *Panouillères et al., 2009*). To help a better intuition for why a visual error model cannot explain learning, please see the simulations of a visual error model in Appendix 1 subsection 1.1 and *Appendix 1—figure 1*.

## Detailed model description

In this section, we give a detailed account of the implementation of the models. We begin by describing the basic processes and the different gains and then proceed to the learning rule and error types.

## Basic processes and gains

We describe the perceived visual position $V_1(n)$ of the pre-saccadic target with the visual gain $\omega_v(n)$:

$$V_1(n) = P_1\omega_v(n), \tag{1}$$

where $n$ is the trial number and $P_1$ is the physical eccentricity of the target (*Figure 2A-1*). The perceived position of the pre-saccadic target on the visual map accurately reflects the physical

eccentricity if $\omega_v(n) = 1$. Different values of $\omega_v(n)$ would reflect mislocalizations that have been observed in localization during fixation (*Müsseler et al., 1999*; *Lappe et al., 2000*; *Stork et al., 2010*) and after saccadic adaptation (*Moidell and Bedell, 1988*; *Collins et al., 2007*; *Hernandez et al., 2008*; *Schnier et al., 2010*; *Zimmerman and Lappe, 2010*; *Gremmler et al., 2014*). In model fitting, $V_1(n)$ corresponds to the pre-saccadic target localization with $\omega_v(n)$ being allowed to plastically learn from error with the learning rate $\alpha_v$.

In the visual-to-motor transformation of the inverse model (*Figure 2A–2*), the visual pre-saccadic target position $V_1(n)$ is mapped onto a motor command $M(n)$ with the motor gain $\omega_m(n)$:

$$M(n) = V_1(n)\omega_m(n) = P_1\omega_v(n)\omega_m(n) \tag{2}$$

If $\omega_m(n) = 1$, the motor command is accurate such that the saccade lands at the spatial position $V_1(n)$. In model fitting, $M(n)$ corresponds to the saccade vector (primary saccade to the target) with $\omega_m(n)$ being allowed to plastically learn from error with the learning rate $\alpha_m$, reflecting learning of the inverse model (the visual-to-motor transform) in response to an assumed motor failure.

The motor command $M(n)$ is copied into $CD_M(n)$:

$$CD_M(n) = M(n) \tag{3}$$

and routed into the CD pathway.

Before saccade onset, the motor-to-visual transformation of the forward dynamics model (*Figures 2A–3*) maps the corollary discharge of the motor command $CD_M(n)$ into the $CD_V(n)$ signal, that is, the computed displacement of visual space, an estimate of the saccade vector in visual coordinates:

$$CD_V(n) = CD_M(n)\omega_{cd}(n) = P_1\omega_v(n)\omega_m(n)\omega_{cd}(n) \tag{4}$$

If the CD gain $\omega_{cd}(n) = 1$, the forward dynamics model is accurate such that $CD_V(n)$ matches the actual saccade vector. To fit the model to a possible imbalance between actual saccade and $CD_V(n)$ induced by learning, the CD gain $\omega_{cd}(n)$ is allowed to plastically learn from error with the learning rate $\alpha_{cd}$. Please note that $\omega_{cd}(n)$ captures the explicit recognition that the motor-efferent $CD_M(n)$ signal needs to be transformed into visuospatial coordinates, that is, $CD_V(n)$, before it can be used by vision.

The computed displacement of visual space $CD_V(n)$, is then routed into a forward outcome model that maps the visual pre-saccadic target position $V_1(n)$ into a prediction about where the post-saccadic target will appear on the retina after the saccade:

$$\hat{V}_2(n) = V_1(n) - CD_V(n) = P_1\omega_v(n)(1 - \omega_m(n)\omega_{cd}(n)) \tag{5}$$

Thus, $\hat{V}_2(n)$ is the predicted visual error (*Figures 2A–4*). It corresponds to the post-saccadic target localization with respect to the post-saccadic gaze position, that is, the gaze position after the primary saccade was performed.

The motor command, when executed, produces the performed saccade vector (*Figures 2A–5*):

$$P_M(n) = M(n) + \epsilon_m(n) = P_1\omega_v(n)\omega_m(n) + \epsilon_m(n) \tag{6}$$

Here, $\epsilon_m(n)$ describes random motor noise in saccade execution. For the model fitting, we will set $\epsilon_m(n) = 0$. Since the models are not fitted to trial-by-trial data but to averages over probe trials during learning, the noise will be canceled out in the fits. Hence, $P_M(n) = M(n)$. We have, however, checked that the basic model performance is robust to the inclusion of motor noise.

In our double-step paradigms, the target is shifted during the saccade. Thus, after saccade landing, the shifted target is displayed with the distance $P_s$ either with respect to the pre-saccadic target position (CTS conditions) or with respect to the post-saccadic gaze position (CVE conditions). Hence, the trans-saccadic target displacement resulting from the imposed target shift and the motor execution noise becomes:

$$CTS : P_d(n) = P_s - \epsilon_m(n) \tag{7}$$

$$CVE : P_d(n) = P_M(n) + P_s - P_1 - \epsilon_m(n) = P_1(\omega_v(n)\omega_m(n) - 1) + P_s \tag{8}$$

Thus, after saccade landing, the post-saccadic target appears at the retinal position (*Figures 2A–6*):

$$V_2(n) = P_1 + P_d(n) + \epsilon_m(n) - P_M(n) = P_1(1 - \omega_v(n)\omega_m(n)) + P_d(n) \tag{9}$$

$V_2(n)$ is the actual visual error that the system receives after the saccade.

## Learning rule

Learning in our model is based on the delta rule that reflects the principle of error-based learning in the sensorimotor system (*Widrow and Hoff, 1960*; *Widrow and Stearns, 1985*). According to the delta rule of error-based learning, the system estimates the gradient of the directional error with respect to every gain of the visuomotor circuitry, thereby deriving whether the error will increase or decrease as the gain will be increased or decreased (*Wolpert et al., 2011*). The system then follows an internal estimate of the gradient to minimize the error as a function of its gains (*Doya, 1999*; *Wolpert et al., 2011*; *Taylor and Ivry, 2014*). Hence, the visuomotor gains

$$\omega(n) = \begin{pmatrix} \omega_v \\ \omega_m \\ \omega_{cd} \end{pmatrix}(n)$$

are adapted after each trial in the direction in which the error $|E|^2$ decreases most rapidly:

$$\omega(n+1) = \omega(n) - \alpha \frac{\partial |E|^2(n)}{\partial \omega(n)} \tag{10}$$

The learning rates

$$\alpha = \begin{pmatrix} \alpha_v & 0 & 0 \\ 0 & \alpha_m & 0 \\ 0 & 0 & \alpha_{cd} \end{pmatrix}$$

determine the speed of learning across trials. If any of the three gains is plastic, its learning rate will be significantly different from zero. If $E(n) = 0$, the system has reached a steady state in which the saccade vector and the visual target representations are stable except for random noise fluctuations.

## Learning from visual prediction error

In the prediction model (*Figures 2A–7*; *Figure 2B*, left), the visuomotor system encodes the error signal as the deviation of the visual post-saccadic target location $V_2(n)$ from the predicted location $\hat{V}_2(n)$:

$$E_{pre}(n) = V_2(n) - \hat{V}_2(n) = P_1(1 + \omega_v(n)(\omega_m(n)(\omega_{cd}(n) - 1) - 1)) + P_d(n) \tag{11}$$

Hence, $E_{pre}(n)$ denotes the amount of visual error that stems from the target step or that was not correctly predicted by the CD gain (*Figure 2A*).

According to the delta rule, the visuomotor gains are adapted after each trial in the direction in which $|E_{pre}|^2$ decreases most rapidly:

$$\omega(n+1) = \omega(n) - \alpha \frac{\partial |E_{pre}|^2(n)}{\partial \omega(n)} \tag{12}$$

$$= \omega(n) - 2\alpha E_{pre}(n)\frac{\partial E_{pre}(n)}{\partial \omega(n)} \tag{13}$$

$$= \omega(n) - 2\alpha E_{pre}(n)\begin{pmatrix} P_1(\omega_m(n)(\omega_{cd}(n) - 1) - 1) \\ P_1\omega_v(n)(\omega_{cd}(n) - 1) \\ P_1\omega_v(n)\omega_m(n) \end{pmatrix} \tag{14}$$

If $E_{pre}(n) = 0$, the post-saccadic target will appear at the retinal position where it was predicted to appear. At that point the system has reached a steady state in which the saccade vector and the visual target representations are stable except for random noise fluctuations.

## Learning from postdictive motor error

In the postdiction model, the visuomotor system combines the visual post-saccadic target position $V_2(n)$ and $CD_V(n)$ to postdictively update the target position in a pre-saccadic frame of reference:

$$\hat{V}_1(n) = V_2(n) + CD_V(n) = P_1(1 + \omega_v(n)\omega_m(n)(\omega_{cd}(n) - 1)) + P_d(n) \tag{15}$$

Analogous to the forward outcome model that predicts the post-saccadic target position (*Figures 2A–4*), we denote this transformation as the backward outcome model as it postdicts the pre-saccadic target position (*Figures 2A–8*; *Figure 2B*, right).

On this basis, the postdictive motor error $E_{post}(n)$ is computed as the error of the motor command with respect to the postdicted pre-saccadic target position (*Figures 2A–9*; *Figure 2B*, right):

$$E_{post}(n) = \hat{V}_1(n) - M(n) = P_1(1 + \omega_v(n)\omega_m(n)(\omega_{cd}(n) - 2)) + P_d(n) \tag{16}$$

The sensorimotor gains adapt after each trial to reduce $|E_{post}|^2$ via the delta rule:

$$\omega(n+1) = \omega(n) - \alpha \frac{\partial |E_{post}|^2(n)}{\partial \omega(n)} \tag{17}$$

$$= \omega(n) - 2\alpha E_{post}(n) \frac{\partial E_{post}(n)}{\partial \omega(n)} \tag{18}$$

$$= \omega(n) - 2\alpha E_{post}(n) \begin{pmatrix} P_1\omega_m(n)(\omega_{cd}(n) - 2) \\ P_1\omega_v(n)(\omega_{cd}(n) - 2) \\ P_1\omega_v(n)\omega_m(n) \end{pmatrix} \tag{19}$$

Hence, $\hat{V}_1(n)$ is a postdictive update of the pre-saccadic desired state to retroactively evaluate the motor command in its native reference frame. This appears appropriate if $CD_V(n)$ is still available after saccade landing (*Cavanaugh et al., 2016*) and the visuomotor system trusts post-saccadic target vision more than the more peripheral pre-saccadic target vision. The postdiction model reaches a steady state if the saccade lands at the postdicted pre-saccadic target position such that $E_{post}(n) = 0$.

## Model fitting and analysis

### Analysis of visuomotor plasticity

Before fitting the models to the data on the basis of the respective error types ($E_{pre}$ or $E_{post}$), we derived the state of $CD_V$ and the visuomotor gains $\omega$ at the five probe blocks based on the basic model rationale. This allowed us to examine our first question 'Which gains are plastic?' independently from our second question 'Which error signal drives this plasticity?'. Based on the pre-saccadic target localization ($V_1$), the saccade vector ($M$) and the post-saccadic target localization with respect to the saccade landing position ($\hat{V}_2$), we derive:

$$CD_V = V_1 - \hat{V}_2 \tag{20}$$

$$\omega_v = \frac{V_1}{P_1} \tag{21}$$

$$\omega_m = \frac{M}{V_1} \tag{22}$$

$$\omega_{cd} = \frac{CD_V}{CD_M} \tag{23}$$

To evaluate the plasticity of each gain, we tested each gain change $\Delta\omega$ from probe block 1 to 5

against zero with a two-sided one-sample t-test. T-tests between gain changes were corrected for the direction of gain change.

## Model fitting and comparison between models

As the error gradient predefines the direction of learning, the models can produce a good fit to the data only if the measured changes in pre-saccadic target localization, saccade vector and post-saccadic target localization occur in the direction of error reduction. If this is not the case, model fitting will not produce a good fit to the data. To test which model produces a good fit and hence, can explain learning, we fitted the models to the individual subject data separately for each condition ($P_1 = 13°$, $P_s = -3°$ for inward, $P_s = +3°$ for outward). This allowed us to compare the models on the basis of the most optimal fits in four learning conditions. Starting from the baseline median saccade vector ($M(1)$), pre-saccadic localization ($V_1(1)$) and post-saccadic localization (with respect to the landing point of the primary saccade, $\hat{V}_2(1)$) of the first probe block, we fitted the learning rates $\alpha$ for which the weighted sum of squared errors (SSE) at trials $n = [1, 71, 141, 211, 281]$ (derived from the medians of the five probe blocks) was minimized:

$$\alpha_{\mathbf{fit}} = argmin \sum_{n=1,71,141,211,281} \eta(n) \; ((V_1(n) - V_{1,predicted}(n))^2 + (M(n) - M_{predicted}(n))^2$$
$$+ (\hat{V}_2(n) - \hat{V}_{2,predicted}(n))^2)$$

Thereby, $\eta(n)$ is the weight according to the number of trials within the probe block to account for the certainty of the data ($\eta(1,281)=1.52$, $\eta(71,141,211)=0.65$). Please note that we did not fit the model to the trial-by-trial data but to the averages of each probe block during learning. This was done because we could obtain pre-saccadic target localization, saccade vector and post-saccadic target localization only from the combination of trials within probe blocks. As localizations and saccade vectors did not systematically change within a probe block, the average of each probe block is considered the best measure of the current state of the visuomotor system during learning.

The lower bound for $\alpha$ was restricted to 0 to ensure that the system learned in gradient direction. The upper bound was set to $9*10^{-5}$ to prevent $M(n)$ from taking a strong exponential shape without an emerging asymptote.

Since both models have the same free parameters (the three visuomotor gains) and hence, exhibit the same amount of model complexity, model selection was based on paired t-tests on the residual standard error $RSE$ between subject prediction and postdiction model fits:

$$RSE = \sqrt{\frac{SSE}{\lambda - 1}} \tag{24}$$

with $\lambda = 15$ as the number of data points used for $RSE$ calculation (five probe blocks each with pre-saccadic localization, saccade vector and post-saccadic localization). Afterwards, we fitted the postdiction model to both conditions, choosing the shared learning rate $\alpha$ that minimizes the SSE summed over the CTS and the CVE condition of each subject.

The error that drives learning is expected to be zero in visuomotor steady states when no systematic changes occur. This should be the case in baseline saccades and when learning has converged. To test error nullification we computed the model's baseline error without target step (CTS with $P_s = 0$), using the gains $\omega$ of the first trial, as well as the final error at the last trial and the percentage of error decline from the first trial (with target step) to the last trial.

We extracted the baseline error $E_{pre}(1)$ and $E_{post}(1)$ if simulated without target step (CTS with $P_s = 0$), the percentage of error decline from trial 1 to trial 281 and the final error $E_{pre}(281)$ and $E_{post}(281)$ of the prediction and the postdiction model fits. Moreover, we computed the $CD_V$ error $CD_V - M$ for baseline saccades and for saccades at the end of the CTS conditions. Since the final error of the model fits cannot become >0 in inward learning and not <0 in outward learning, statistical tests against zero were performed on the $E_{post}(281)$ and $E_{pre}(281)$ errors derived directly from the data.

## Stability analysis

We performed stability analysis of the postdiction model to examine the steady states to which the visuomotor gains can converge. The trial-by-trial gain change of the postdiction model is described by a system of three-dimensional nonlinear partial differential equations:

$$\Delta\omega(n) = -\alpha \frac{\partial |E_{post}|^2(n)}{\partial \omega(n)} \tag{25}$$

We set $\Delta\omega(n) = 0$ to extract the fixed points to which the postdiction model can converge. The stability of the fixed points was evaluated based on the trace $\tau$, the determinant $det$ and $\tau^2 - 4det$ extracted from the Jacobian matrix of $\Delta\omega(n)$

$$J(\Delta\omega(n)) = \begin{pmatrix} \dfrac{\partial \Delta\omega_v(n)}{\partial \omega_v(n)} & \dfrac{\partial \Delta\omega_v(n)}{\partial \omega_m(n)} & \dfrac{\partial \Delta\omega_v(n)}{\partial \omega_{cd}(n)} \\ \dfrac{\partial \Delta\omega_m(n)}{\partial \omega_v(n)} & \dfrac{\partial \Delta\omega_m(n)}{\partial \omega_m(n)} & \dfrac{\partial \Delta\omega_m(n)}{\partial \omega_{cd}(n)} \\ \dfrac{\partial \Delta\omega_{cd}(n)}{\partial \omega_v(n)} & \dfrac{\partial \Delta\omega_{cd}(n)}{\partial \omega_m(n)} & \dfrac{\partial \Delta\omega_{cd}(n)}{\partial \omega_{cd}(n)} \end{pmatrix} \tag{26}$$

in which the fixed point equations were inserted.

## Statistical analysis

A mixed analysis of variance (ANOVA) was computed each for the saccade vector, pre- and post-saccadic localization changes (learning direction inward/outward as between-subject factor, paradigm CTS/CVE as a within-subject factor, Greenhouse-Geisser corrected). As a mixed ANOVA on gain changes (learning direction as between-subject factor, paradigm and gain type as within-subject factors) revealed significant main effects of paradigm and gain as well as a significant interaction between direction and paradigm, we performed a repeated measures one-way ANOVA over gain change for each learning condition. To compare two groups of data or fitted parameters or to test one group of data against zero, two-sided t-tests or alternatively, two-sided Wilcoxon signed-rank tests were applied if normal distribution was violated. Tests were performed with a significance level of 0.05 except for Bonferroni-corrected post-hoc t-tests.

## Results

### Learning induces changes in saccade amplitude, pre- and post-saccadic localization

We first wanted to determine the states of the different aspects of the visuomotor transform during learning from the experimental data irrespective of the error model. We collected data on the saccade vector and on the pre- and post-saccadic localization of visual targets during a double-step task with four different target shift conditions. In the $CTS_{in}$ condition the target stepped a constant 3° against the saccade direction in each trial, leading to a constant inward target shift. In the $CTS_{out}$ condition, the target stepped a constant 3° in the saccade direction in each trial, leading to a constant outward target shift. In the CVE conditions, the target stepped to a location 3° from the landing point of the saccade, either against saccade direction in the $CVE_{in}$ condition or in saccade direction in the $CVE_{out}$ condition, each time leading to a constant visual error (inward and outward, respectively). In each of the four conditions, we calculated the CD gain according to the basic model rationale from the difference between post-saccadic localization, pre-saccadic localization and the saccade vector.

*Figure 3A* shows the mean data across subjects for all four conditions and *Figure 3B* shows the data of an example subject in the $CTS_{out}$ condition (for all individual subject data see *Appendix 1—figures 3–6*). Consistent with previous studies we found large changes in saccade vector (*McLaughlin, 1967*; *Miller et al., 1981*; *Wallman and Fuchs, 1998*; *Bahcall and Kowler, 1999*; *Panouillères et al., 2009*; *Ethier et al., 2008a*; *Havermann and Lappe, 2010*) and post-saccadic localization (*Bahcall and Kowler, 1999*; *Collins et al., 2007*; *Zimmermann and Lappe, 2009*; *Schnier et al., 2010*; *Klingenhoefer and Bremmer, 2011*) that were significant in all learning

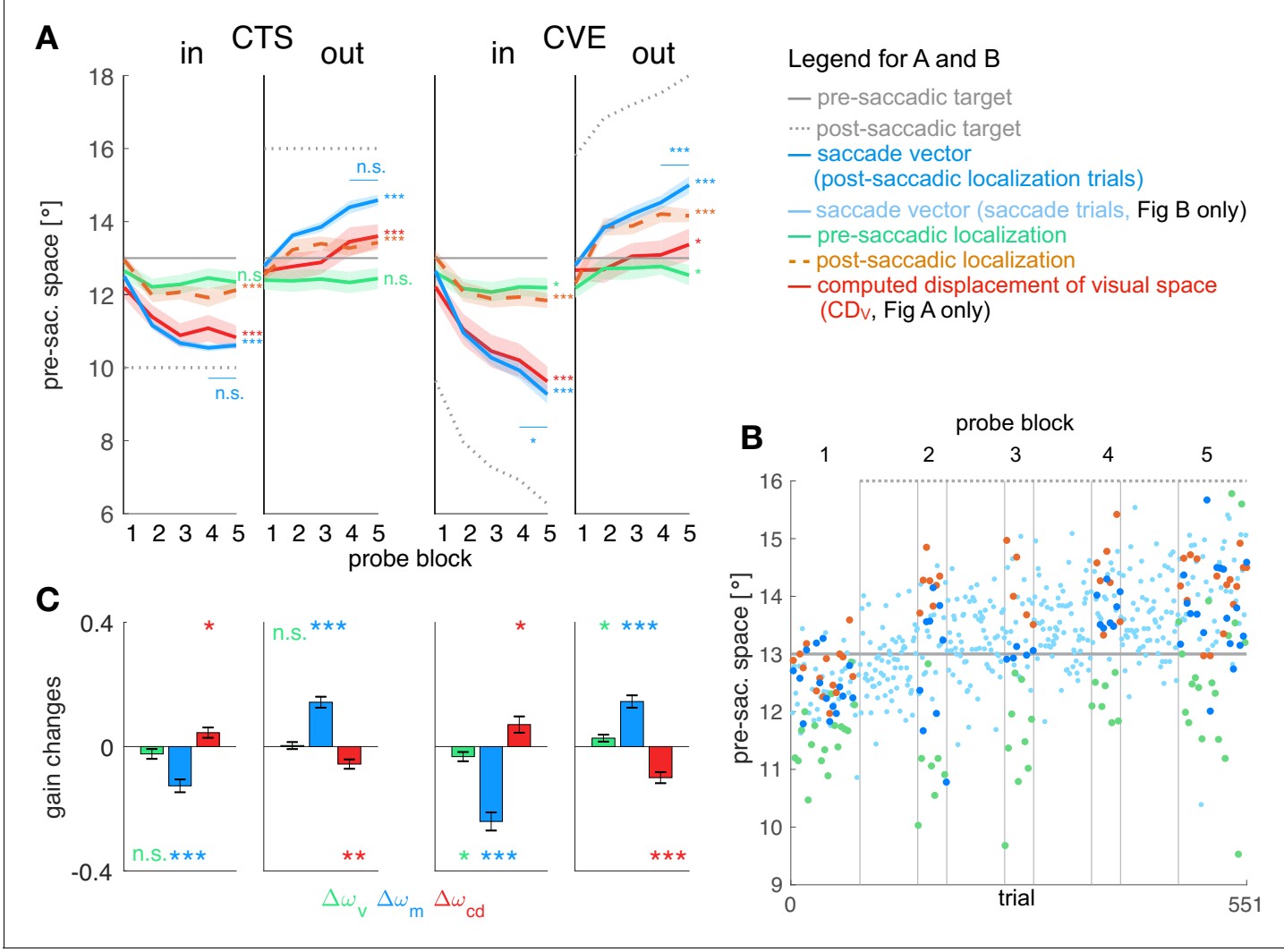

**Figure 3.** Experimental data and gain changes. (A) Experimental data averaged across subjects. Each panel shows saccade vectors, pre- and post-saccadic localizations and $CD_V$ of the five probe blocks for a specific learning condition. Error bars indicate standard error of the mean. Asterisks at the right edge of each panel indicate significant change from probe block 1 to 5 with ***p<0.001, **p<0.01, *p<0.05 and n.s. p≥0.05. Blue asterisks within the panel indicate significant change from probe block 4 to 5 showing that learning was completed at the end of the CTS conditions but still in progress at the end of the CVE conditions. (B) Experimental data of an example subject for the $CTS_{out}$ condition. Saccade vectors, pre- and post-saccadic localizations were measured within five probe blocks across learning. As saccade vectors needed to be related to post-saccadic localizations, only the saccade vectors of the post-saccadic localization trials (dark blue) were used for the analysis. (C) Gain changes from the first to the last trial for the four learning conditions, averaged across subjects. Asterisks indicate significant difference from zero with ***p<0.001, **p<0.01, *p<0.05 and n.s. p≥0.05.

conditions (p<0.001, see *Table 1* for detailed statistical analysis). As expected from other studies, these changes were larger in the CVE paradigms than in the CTS paradigms (main effect of paradigm $F_{1,31}$ = 31.95, p<0.001 for saccade vectors, $F_{1,31}$ = 24.60, p<0.001 for post-saccadic localizations, both post-hoc t-tests p<0.001, see *Table 1*, *Robinson et al., 2003*; *Havermann and Lappe, 2010*; *Zimmerman and Lappe, 2010*). Also in accordance with previous work, the changes in pre-saccadic localization were small, but were significantly different from zero in the CVE conditions (p ≤ 0.046, *Moidell and Bedell, 1988*; *Collins et al., 2007*; *Hernandez et al., 2008*; *Schnier et al., 2010*; *Zimmerman and Lappe, 2010*; *Gremmler et al., 2014*).

In both CTS conditions, the saccade vector at the end of the session appears to be converged to a new steady state (saccade vector from probe block 4 to 5 vs. zero, $CTS_{in}t_{16}$ = 0.787, p=0.443; $CTS_{out}$ $t_{17}$ = 1.75, p=0.097). In the CVE conditions, in contrast, the saccade vector did not appear to

reach a steady state and might further adapt with more trials ($CVE_{in}$ $t_{16} = -3.67$, p=0.002; $CVE_{out}$ $t_{17} = 4.70$, p<0.001). For model fitting, this means that the error signal should be nullified at the end of the CTS conditions but not at the end of the CVE conditions as, here, learning is still in progress.

## Plasticity is reflected in gains of visual target percept, motor command and $CD_V$

Before fitting the models to the data on the basis of the respective learning rules ($E_{pre}$ or $E_{post}$ learning), we derived the gain changes $\Delta\omega$ and deduced the state of $CD_V$ at the five probe blocks based on the basic model rationale. This allowed us to examine which visuomotor gains are plastic (first question, basic model) independently from the question of which error signals can explain this plasticity (second question, model fitting to the learning rules). The time course of $CD_V$ is presented in the red lines in *Figure 3A*. The gain changes $\Delta\omega$ of the three signals (vision, motor and CD gain) are shown in *Figure 3C*. These visuomotor gain changes describe the plasticity of the respective signal with respect to the other signals. Hence, if a signal is plastic, the respective gain change should systematically differ from zero across subjects.

In the case of the visual gain $\omega_v$, changes directly result from the pre-saccadic target localization. Hence, the visual gain changed in the direction of the target step which was significant in the CVE conditions ($CVE_{in}$ $t_{16} = -2.19$, p=0.046; $CVE_{out}$ $t_{17} = 2.44$, p=0.026) but not in the CTS conditions ($CTS_{in}$ $t_{16} = -1.49$, p=0.159; $CTS_{out}$ $t_{17} = 0.30$, p=0.766). For the CVE conditions, this suggests that the visual gain is adapted in response to error, as the error might result from a deficient pre-saccadic target localization on the spatial map.

In the case of the motor gain $\omega_m$, changes result from the saccade vector with respect to the pre-saccadic target localization. In all conditions, the motor gain significantly changed during learning ($CTS_{in}$ $t_{16} = -6.10$, p<0.001; $CTS_{out}$ $t_{17} = 8.08$, p<0.001; $CVE_{in}$ $t_{16} = -8.24$, p<0.001; $CVE_{out}$ $t_{17} = 7.28$, p<0.001). This suggests that the inverse model adapts its transformation from visual target percept to the motor command in response to a presumed change in eye dynamics.

A key question of our study is how the CD gain of the forward dynamics model develops during learning. It has been argued that $CD_V$ remains fixed at baseline level such that the visual system is completely unaware of the ongoing motor changes (*Bahcall and Kowler, 1999*) or, at least, that the $CD_V$ signal might not correctly reflect the motor changes during learning (*Collins et al., 2009*). In this case, $CD_V$ would deviate from the performed saccade vector and, in turn, produce post-saccadic mislocalization. In contrast, if the CD gain correctly reflects the motor changes, it should accurately

**Table 1.** Analysis of the experimental data.

Here, we report mean and standard deviation of changes in saccade vector, pre- and post-saccadic localization from probe block 1 to 5. T-values are derived from two-sided t-tests against zero. F-values are derived from 2 × 2 mixed ANOVAs for saccade vector, pre- and post-saccadic localization changes (corrected for direction) with paradigm (CTS/CVE) as within-subject factor and direction (in/out) als between-subject factor. Changes in saccade vector were significant in all conditions but higher for inward than outward learning (within the CVE conditions, post-hoc t-test $t_{33} = 3.68$, p<0.001) and higher for CVE than CTS learning (within the inward conditions, post-hoc t-test $t_{32} = 4.94$, p<0.001, $CVE_{in}$ vs. $CTS_{out}$ $t_{33} = -5.13$, p<0.001, all other post-hoc tests p≥0.126, bonferroni-corrected significance level 0.008). Changes in post-saccadic localization were significant in all conditions but higher for CVE than CTS learning (within the outward conditions, post-hoc t-test $t_{34} = -4.23$, p<0.001, $CVE_{out}$ vs. $CTS_{in}$ $t_{33} = -4.48$, p<0.001, $CVE_{out}$ vs. $CVE_{in}$ $t_{33} = -2.42$, p=0.021, all other post-hoc tests p≥0.163, bonferroni-corrected significance level 0.008). Changes in pre-saccadic localization were small but significant in the CVE conditions.

| | Pre-saccadic localization | Saccade vector | Post-saccadic localization |
|---|---|---|---|
| $CTS_{in}$ | $-0.31 \pm 0.85°$, $t_{16} = -1.48$, p = 0.159 | $-1.89 \pm 0.61°$, $t_{16} = -12.69$, p < 0.001*** | $-0.82 \pm 0.65°$, $t_{16} = -5.24$, p < 0.001*** |
| $CTS_{out}$ | $+0.04 \pm 0.62°$, $t_{17} = 0.30$, p = 0.766 | $+1.79 \pm 0.71°$, $t_{17} = 10.80$, p < 0.001*** | $+0.88 \pm 0.67°$, $t_{17} = 5.57$, p < 0.001*** |
| $CVE_{in}$ | $-0.42 \pm 0.80°$, $t_{16} = -2.17$, p = 0.046* | $-3.37 \pm 1.08°$, $t_{16} = -12.88$, p < 0.001*** | $-1.20 \pm 0.87°$, $t_{16} = -5.66$, p < 0.001*** |
| $CVE_{out}$ | $+0.35 \pm 0.62°$, $t_{17} = 2.44$, p = 0.026* | $+2.19 \pm 0.80°$, $t_{17} = 11.59$, p < 0.001*** | $+1.85 \pm 0.70°$, $t_{17} = 11.17$, p < 0.001*** |
| paradigm | $F_{1,31} = 3.48$, p = 0.071 | $F_{1,31} = 31.95$, p < 0.001*** | $F_{1,31} = 24.60$, p < 0.001*** |
| direction | $F_{1,31} = 0.56$, p = 0.462 | $F_{1,31} = 8.24$, p = 0.007** | $F_{1,31} = 2.94$, p = 0.096 |
| interaction | $F_{1,31} = 0.76$, p = 0.390 | $F_{1,31} = 10.75$, p = 0.002** | $F_{1,31} = 4.75$, p = 0.036* |

describe the saccade vector during learning, accurately compute the displacement of visual space for this saccade, and produce no post-saccadic mislocalization. Note that, in the mathematical formalization of our model, $\omega_{cd} = 1$ means that the forward dynamics model transforms the corollary discharge of the motor command $CD_M$ into an accurate computed displacement of visual space $CD_V$. Hence a deviation between $CD_V$ and the performed saccade vector, as proposed by *Bahcall and Kowler, 1999* and *Collins et al., 2009*, formally corresponds to a change of CD gain in our model.

*Figure 3C* shows the change in the CD gain $\Delta\omega_{cd}$, revealing significant differences from zero in all learning conditions (CTS$_{in}$ $t_{16}$ = 2.72, p=0.015; CTS$_{out}$ $t_{17}$ = −3.70, p=0.001; CVE$_{in}$ $t_{16}$ = 2.70, p=0.016; CVE$_{out}$ $t_{17}$ = −5.61, p<0.001). This suggests that the $CD_V$ signal does not correctly reflect the adapting saccade vector during learning (*Bahcall and Kowler, 1999*; *Collins et al., 2009*). With respect to the visuomotor transformations in the brain, this means that the forward dynamics model adapts its transformation from $CD_M$ to $CD_V$, that is, the CD gain. The changes in CD gain occur in the opposite direction of motor gain changes (*Figure 3C*), reflecting the opposing dynamics between inverse and forward dynamics model as if the system associates the error with a change in eye dynamics.

*Figure 3A* also illustrates how much the computed displacement of visual space $CD_V$ (red line) dissociates from the actual saccade vector (blue line) during learning (remember that the dissociation between pre- and post-saccadic target localization determines the size of the $CD_V$ with respect to the actual saccade vector). The CD gain does not correctly reflect the saccade vector since $\Delta\omega_{cd}$ significantly differs from zero. However, it also does not remain fixed at the baseline level during learning ($\Delta CD_V$ significantly differs from zero, CTS$_{in}$ $t_{16}$ = −5.99, p<0.001; CTS$_{out}$ $t_{17}$ = 3.99, p<0.001; CVE$_{in}$ $t_{16}$ = −8.94, p<0.001; CVE$_{out}$ $t_{17}$ = 2.39, p=0.029). Instead, the CD gain reflects saccade changes in the direction of the target step but underestimates the size of the saccade change.

Finally, we wanted to compare plasticity between gains in order to determine which signals provide strong or weak contributions to the learning effect. We performed a repeated measures one-way ANOVA for each condition, revealing a main effect of gain type ($\omega_v$, $\omega_m$ or $\omega_{cd}$) in each condition ($F_{2,32}$ = 7.14, p=0.003 for CTS$_{in}$, $F_{2,34}$ = 19.81, p<0.001 for CTS$_{out}$, $F_{2,32}$ = 19.66, p<0.001 for CVE$_{in}$ and $F_{2,34}$ = 10.98, p<0.001 for CVE$_{out}$). In all conditions, the motor gain change $\Delta\omega_m$ was larger than the CD gain change $\Delta\omega_{cd}$ (post-hoc t-tests p≤0.019) which was larger than the visual gain change $\Delta\omega_v$ in the outward conditions (post-hoc t-tests p≤0.044, corrected for the direction of gain change).

To summarize the plastic changes in the gains before we turn to the modeling, we found most plasticity within the saccadic motor command and within the $CD_V$ signal, and small but significant plasticity within the pre-saccadic target percept in the CVE conditions. Illustrations of the role of each individual gain in the learning process are presented in Appendix 1 subsection 1.2 and *Appendix 1—figure 2*.

## Postdictive motor error drives learning

After revealing that all three visuomotor gains are plastic, we examined which learning rule can explain this plasticity. We compare the minimization of visual prediction error $E_{pre}$ with the minimization of postdictive motor error $E_{post}$. We fitted both models to the data of the pre-saccadic target localizations ($V_1$), saccade vectors ($M$) and post-saccadic target localizations (with respect to the saccade landing position, $\hat{V}_2$). A model will only fit well if error reduction is consistent with the measured changes in pre-saccadic target localization, saccade vector and post-saccadic target localization.

*Figure 4* shows the fits of the prediction model (A) and the postdiction model (B) to the data. Please note that the lines represent the model fit and hence, appear smooth compared to the mean over subjects represented by the lines in *Figure 3A*. In *Figure 4*, the shaded areas in the background represent the measured data of pre-saccadic target localization (green, fitted by $V_1$), saccade vector (blue, fitted by $M$), and post-saccadic target localization (dashed orange line, second row, fitted by $\hat{V}_2$).

The fit of the prediction model (*Figure 4A*) was not able to capture the data, neither for the motor performance (blue lines) nor for the observed perceptual effects (green lines in top panels, orange dashed lines in second panels). The reason for the failure of the prediction model can directly be seen in the data. The visual prediction error is the difference between the predicted post-saccadic target position $\hat{V}_2$ (dashed orange line in second panel) and the actual post-saccadic target

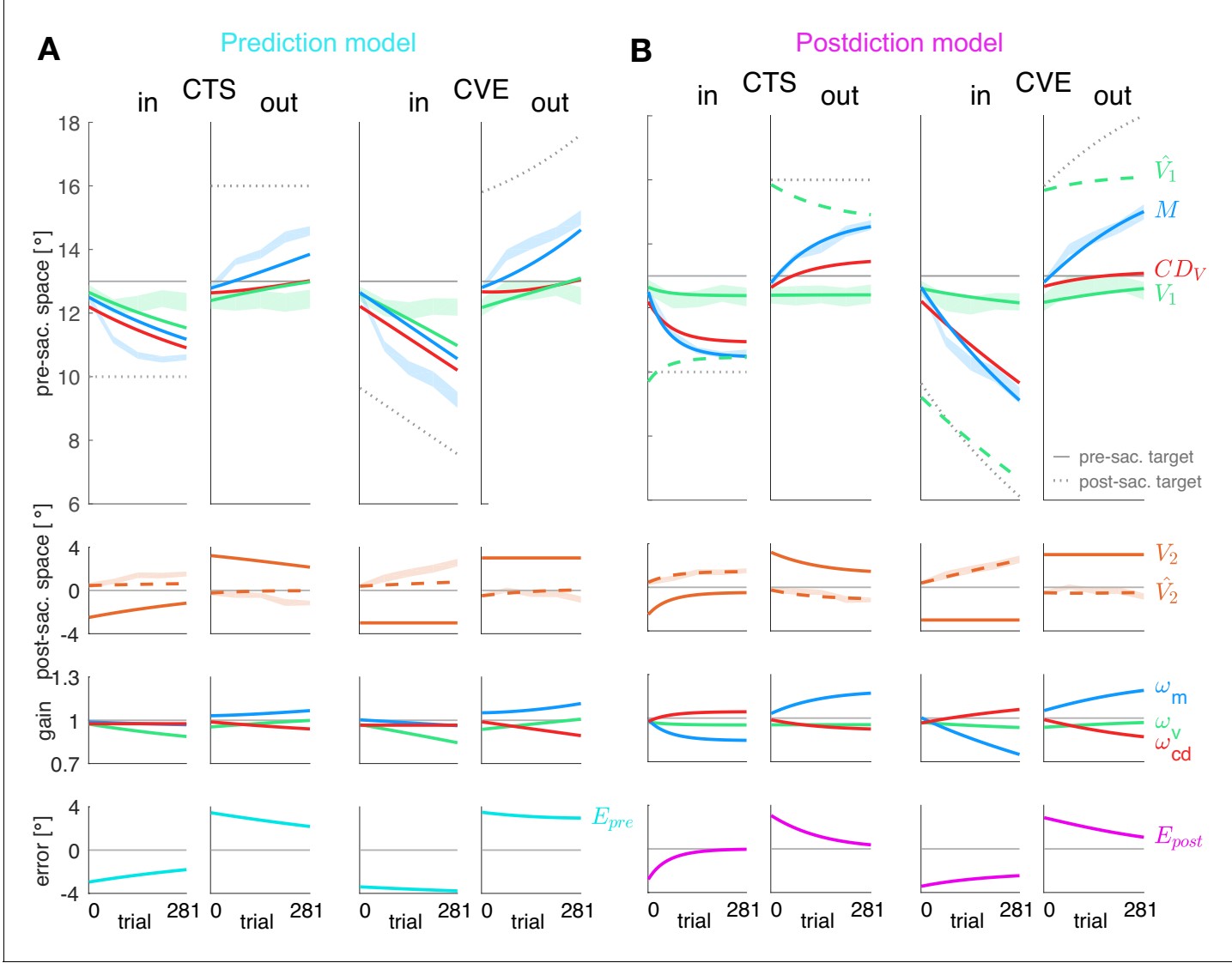

**Figure 4.** Prediction model fits and postdiction model fits to the experimental data. Each column shows the fit for a specific learning condition. Fits are shown by lines. Data (subject means ± standard error) are shown by shaded areas. Please note that due to the model fit, the lines appear smooth compared to the mean over subjects represented by the lines in *Figure 3A*. First row: Visual pre-saccadic target $V_1$ (fitted to pre-saccadic retinal localizations, green shade), motor command $M$ (fitted to saccade vectors, blue shade), computed displacement of visual space $CD_V$, postdicted pre-saccadic target $\hat{V}_1$ in the postdiction model. Second row: Predicted post-saccadic target $\hat{V}_2$ (fitted to post-saccadic retinal localizations, orange shade), visual post-saccadic target $V_2$. Third row: Visual gain $\omega_v$, motor gain $\omega_m$, CD gain $\omega_{cd}$. (A) Last row: Visual prediction error $E_{pre} = V_2 - \hat{V}_2$. To nullify $E_{pre}$, the model requires $CD_V$ to learn in the opposite direction of $M$. As this is not in line with the data, the prediction model does not adequately fit the data. Fitted learning rates are: $CTS_{in}$ $\alpha$ = (4.9*10$^{-6}$; 5.0*10$^{-5}$; 3.8*10$^{-17}$), $CVE_{in}$ $\alpha$ = (4.6*10$^{-6}$; 5.0*10$^{-5}$; 3.4*10$^{-17}$), $CTS_{out}$ $\alpha$ = (2.1*10$^{-6}$; 5.0*10$^{-5}$; 2.3*10$^{-6}$), $CVE_{out}$ $\alpha$ = (2.9*10$^{-6}$; 5.0*10$^{-5}$; 4.0*10$^{-6}$). (B) Last row: Postdictive motor error $E_{post} = \hat{V}_1 - M$. At the end of the CTS conditions, $E_{post}(281) \approx 0$ so that the system is appears to be converged to a steady state while at the end of the CVE conditions learning is still in progress. Fitted learning rates are: $CTS_{in}$ $\alpha$ = (5.2*10$^{-6}$; 3.5*10$^{-5}$; 1.8*10$^{-5}$), $CVE_{in}$ $\alpha$ = (1.9*10$^{-6}$; 1.3*10$^{-5}$; 5.4*10$^{-6}$), $CTS_{out}$ $\alpha$ = (8.4*10$^{-8}$; 1.5*10$^{-5}$; 6.3*10$^{-6}$), $CVE_{out}$ $\alpha$ = (2.0*10$^{-6}$; 9.9*10$^{-6}$; 8.0*10$^{-6}$).

position $V_2$ (continuous orange line in second panel). Clearly, these two lines do not converge and often the predicted post-saccadic target position is in the opposite direction from the actual post-saccadic target position. This is even true at the end of the $CTS_{in}$ condition when the saccade vector data (blue dashed area in top row) are in an asymptotic steady state and the error that drives motor learning should be zero. Clearly, this is not the case for the visual prediction error (bottom row). This

shows that the measured changes in pre-saccadic target localization, saccade vector and post-saccadic target localization are not consistent with minimization of visual prediction error $E_{pre}$.

The postdiction model (*Figure 4B*) fits well to the data, reflecting the respective visuomotor gain changes (third row) and the reduction of the postdictive motor error $E_{post}$ (bottom row). The postdictive motor error is the difference between the postdicted target position $\hat{V}_1$ (dashed green line in top row) and the motor command $M$ (blue line in top row). The postdictive motor error reduces as these lines converge closer together. The convergence is faster in the CTS conditions, consistent with the data.

*Figure 5A* presents the residual standard error for both models. Since both models have the exact same free parameters, that is, the three learning rates for the gains, the residual standard error allows a direct comparison of the fit quality. In all conditions, residual standard errors were smaller for the postdiction model fit than for the prediction model fit (CTS$_{in}$ $t_{16}$ = 5.30, p<0.001; CTS$_{out}$ $t_{17}$ = 3.76, p=0.002; CVE$_{in}$ $t_{16}$ = 5.84, p<0.001; CVE$_{out}$ $t_{17}$ = 3.42, p=0.003). After model comparison on the basis of the separately fitted conditions, we also fitted the learning rates $\alpha_v$, $\alpha_m$ and $\alpha_{cd}$ that minimize SSE summed over the CTS and the CVE condition of each subject. Here, again, the residual standard error was smaller in the postdiction model fit than in the prediction model fit in all conditions (p $\leq$ 0.015). For the postdiction model, the residual standard error when fitting shared learning rates was 0.54 ± 0.19° for CTS$_{in}$, 0.45 ± 0.12° for CTS$_{out}$, 0.58 ± 0.18° for CVE$_{in}$ and 0.49 ± 0.14° for CVE$_{out}$.

In sum, the postdiction model well describes learning of the saccade vector and the pre-and post-saccadic localization data, including a dissociation of $CD_V$ from the saccade in the sense of an underestimation of saccade changes in learning direction. In contrast, the prediction model fails to capture perceptual and saccadic data.

## Postdictive motor error explains visuomotor steady states

In this section, we derive some essential properties of the steady states of the postdiction model that relate to properties of saccades, saccadic adaptation, and trans-saccadic perception.

In designing our model, we aimed to find a learning rule that can explain continuous calibration of motor performance and spatial perception without the need for a pre-defined baseline state that restrains learning. Instead, we propose that visuomotor steady states are achieved when the error signal is nullified, that is, when no systematic changes occur except for random noise fluctuations. This should first be the case in baseline saccades (without a peri-saccadic target step) and, second, when plasticity has converged after learning with a peri-saccadic target step. We analyzed whether the postdictive motor error fulfils this criterion. First, we examined whether the postdictive motor error is nullified at baseline and in converged adapted steady states. Second, we investigated via dynamical systems stability analysis how these steady states are formed.

Subject means of baseline error and medians of final error and error decline are depicted in *Figure 5B–D* (median is depicted for skewed distributions). Baseline $E_{post}$ was nullified in all learning conditions (p≥0.323). By contrast, baseline $E_{pre}$ was not nullified in the outward conditions (CTS$_{out}$ $t_{17}$ = 2.73, p=0.014; CVE$_{out}$ $t_{17}$ = 4.25, p<0.001). Hence, the postdiction model can explain the steady state in baseline saccades.

Especially interesting is the end state in the CTS conditions for which the saccade vector data show asymptotic convergence toward the end of the measurement (*Figure 3A*, CTS$_{in}$ and CTS$_{out}$). This means that learning had reached a steady state in the CTS conditions. Consistent with this, final $E_{post}$ was nullified in these conditions (CTS$_{in}$ $t_{16}$ = −1.36, p=0.193; CTS$_{out}$ $t_{17}$ = 1.51, p=0.150; *Figure 5C*). This was different in the CVE conditions in which learning was still in progress at the end of the measurement (*Figure 3A* CVE$_{in}$ and CVE$_{out}$). Consistent with this observation, $E_{post}$ was still different from zero at that time (CVE$_{in}$ $t_{16}$ = −9.31, p<0.001; CVE$_{out}$ $t_{17}$ 4.78, p<0.001). The prediction model produced final $E_{pre}$ values that were not nullified in any condition (p<0.001), inconsistent with the CTS data. Moreover, the absolute final $E_{post}$ was consistently smaller than the absolute final $E_{pre}$ across all learning conditions (all conditions p<0.001). As the final error of the model fits cannot become >0 in inward learning or <0 in outward learning, statistical tests against zero were performed on the final $E_{pre}$ and $E_{post}$ derived directly from the data of the last probe block. $E_{post}$ was reduced more than $E_{pre}$ across all learning conditions (Wilcoxon signed-rank tests, all conditions

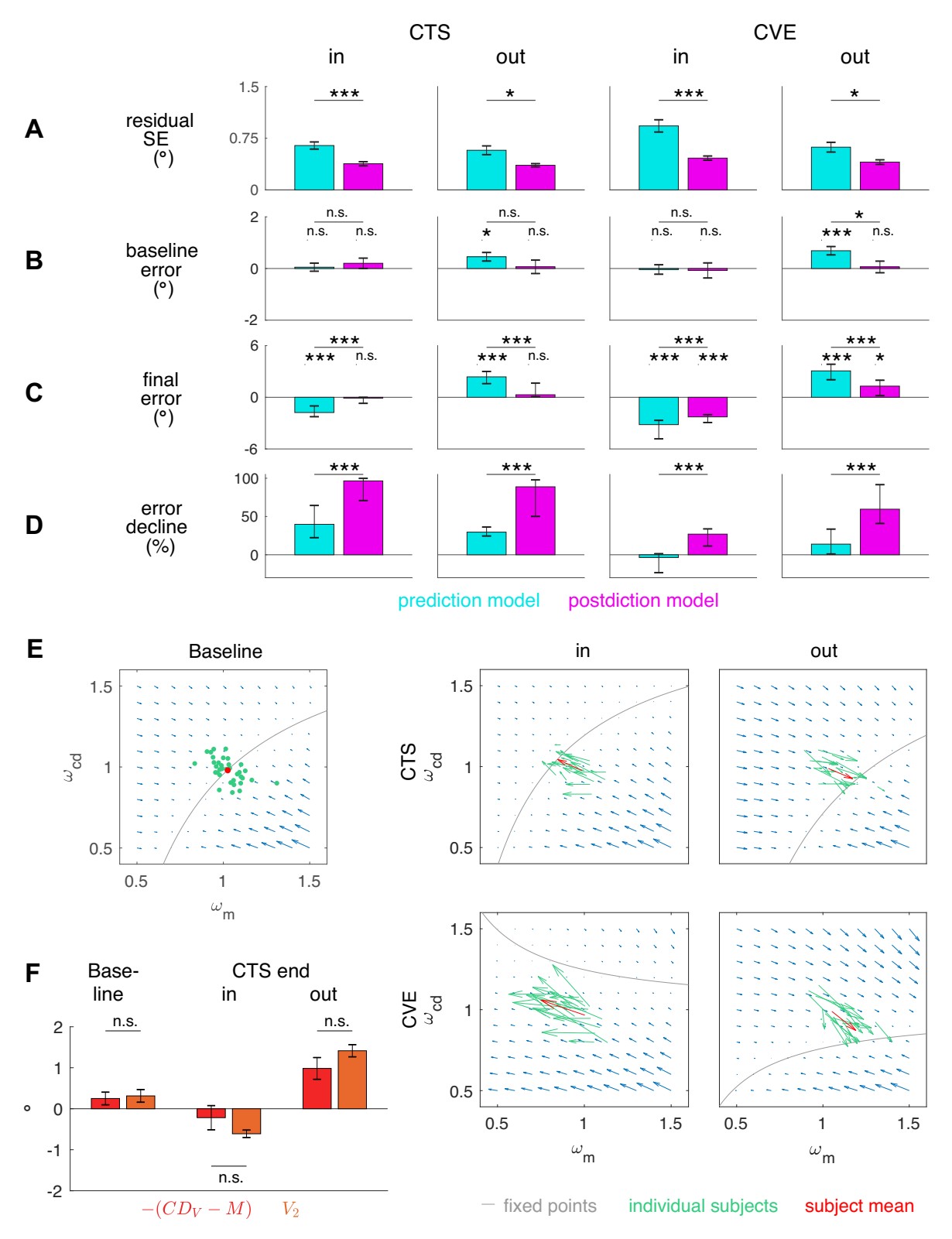

**Figure 5.** Residual standard error and visuomotor steady states. (A) Residual standard error of the prediction and the postdiction model fit (subject means ± standard error). (B) Baseline $E_{pre}$ and baseline $E_{post}$ if no target step occurs (CTS with $P_s$ = 0, subject means ± standard error). The baseline error should be close to zero as the system is assumed to be in a steady state. (C) Final $E_{pre}$ of the prediction model fit and final $E_{post}$ of the postdiction model fit (subject medians with 25% and 75% quantiles). The final error should be close to zero if the system converged to a new steady state. (D)

*Figure 5 continued on next page*

Figure 5 continued

Percentage of $E_{pre}$ decline of the prediction model fit and $E_{post}$ decline of the postdiction model fit from the first trial (including target step) to the last trial (subject medians with 25% and 75% quantiles). Black asterisks indicate significant difference between prediction and postdiction model values, colored asterisks for baseline and final error indicate significant difference from zero. (E) Vector fields depict non-isolated fixed points for the baseline situation without target step and the four learning conditions (in the $\omega_m - \omega_{cd}$ plane for simplicity). Subject median $\omega_v$ of the first trial (for baseline) and the last trial (for learning in the respective condition) and median $\alpha$ from the respective learning conditions were chosen to draw the vector field. Subjects were at visuomotor steady state in the baseline and learned in the direction of the shifted fixed points during learning. (F) The $CD_V$ error determines how much visual endpoint error (visual post-saccadic target eccentricity) is left in the baseline adapted steady state of the CTS conditions (subject means ± standard error). Please note that $-(CD_V - M)$, not $+(CD_V - M)$, is depicted for easier comparison to $V_2$. Black asterisks indicate significant difference from zero with ***p<0.001, **p<0.01, *p<0.05 and n.s. p≥0.05.

p<0.001; *Figure 5D*). In sum, only the postdiction model explains visuomotor steady states in baseline saccades and after learning has converged.

Because our model contains learning of three different gains, it is possible that different combinations of theses gains comprise steady states. To examine the steady states to which the visuomotor gains can converge, we performed stability analysis of the postdiction model. The trial-by-trial change of the three visuomotor gains ends with $\Delta\omega(n) = 0$ when $E_{post}(n) = 0$. Assuming $\omega_v(n) \neq 0$, we derived a plane of stable fixed points for the CTS conditions with:

$$\omega_m(n) = -\frac{P_1 + P_s}{P_1 \omega_v(n)(\omega_{cd}(n) - 2)} \tag{27}$$

For the CVE conditions, we derived a plane of stable fixed points with:

$$\omega_m(n) = -\frac{P_s}{P_1 \omega_v(n)(\omega_{cd}(n) - 1)} \tag{28}$$

*Figure 5E* depicts the steady state curves along with vector fields of the gradient descent learning directions for baseline saccades (without target step) and for the four learning conditions with target step (on a $\omega_m - \omega_{cd}$ plane for simplicity). As described by the steady state equations above, the pre-saccadic target position $P_1$ and the target step $P_s$ (with respect to $P_1$ in the CTS conditions and with respect to the saccade landing position $P_M$ in the CVE conditions) define a plane of visuomotor gains at which the system is at steady state. With $P_s = 0$ this is the case in the baseline situation (*Figure 5E*, CTS with $P_s = 0$). If a target step is introduced, the plane of steady states is shifted in visuomotor gain space depending on $P_s$. Then, the visuomotor gains adapt to one of these new steady states during learning, depending on the initial condition of visuomotor gains (previous steady state, that is, the baseline of our experiment) and the learning rates $\alpha$ which define the skewness of the depicted vector field. *Figure 5E* shows that learning is close to steady state at the end of $CVE_{out}$ but is expected to progress further in $CVE_{in}$.

## $CD_V$ hypometry explains saccade hypometry

A long-standing question in saccade research is why saccades usually undershoot their target by 5–10% (*Robinson, 1973*; *Henson, 1979*; *Becker, 1989*) and why saccadic adaptation does not fully compensate for peri-saccadic target steps (*Deubel et al., 1986*; *Straube et al., 1997*; *Wallman and Fuchs, 1998*; *Noto et al., 1999*). We can also phrase this question as: Why does the system accept a remaining visual endpoint error in baseline and adapted saccades? In the postdiction model, the steady states are characterized by $E_{post}(n) = \hat{V}_1(n) - M(n) = V_2(n) + CD_V(n) - M(n) = 0$. From this, it follows that $V_2(n) = M(n) - CD_V(n)$ in steady state. Thus, the amount of visual endpoint error that the visuomotor system accepts with respect to the post-saccadic target is determined by the accuracy of $CD_V$ with respect to the actual saccade. In other words, the accuracy of the $CD_V$ signal shapes the accuracy of the saccade.

*Figure 5F* compares the visual endpoint error $V_2$ to the $CD_V$ error in the baseline situation and at the end of the CTS conditions. For easier comparison, $-(CD_V - M)$ is plotted instead of $CD_V - M$ since $CD_V - M$ is in the opposite direction of $V_2$. In baseline saccades, $CD_V$ is nearly accurate but shows the tendency to underestimate the saccade (t-test of $\omega_{cd}(1)$ against 1, $t_{34} = -1.67$, p=0.052) which, by means of the above, predicts the amount of baseline target undershoot (t-test $V_2(1)$ vs.

$-(CD_V(1) - M(1)), t_{34} = 4.12$, p=0.678). Hence, $CD_V$ hypometry (with respect to the saccade) prompts the visuomotor system to stabilize with saccade hypometry (with respect to the target) to keep postdictive motor error nullified. This provides a new explanation for the long-known saccade undershoot based on visuomotor learning.

### A dissociation between saccade and $CD_V$ explains incomplete saccade learning from artificial target steps

The relation between motor performance and $CD_V$ signaling also explains the second part of the above question — why saccadic adaptation does not fully compensate for peri-saccadic target steps (*Deubel et al., 1986*; *Straube et al., 1997*; *Wallman and Fuchs, 1998*; *Noto et al., 1999*). *Figure 5F* depicts the visual endpoint error $V_2$ and the $CD_V$ error at the end of the CTS conditions, that is, when the system has reached a steady state. Like in the baseline situation, $V_2$ and $CD_V$ error did not significantly differ at that point (CTS$_{in}$ condition: $t_{16} = -1.36$, p=0.194; CTS$_{out}$ condition: $t_{17} = 1.51$, p=0.150). We can interpret this in the notion of the postdiction model as follows: To minimize the postdictive motor error, visual and motor gains adapt such that the saccade lands successively closer to the post-saccadic target, thereby automatically reducing the amount of visual endpoint error $V_2$ after the saccade. Simultaneously, plasticity of the CD gain in the forward dynamics model dissociates the $CD_V$ signal from the actual saccade vector, depending on the direction of the target step. Learning converges when the current dissociation of the $CD_V$ signal from the saccade corresponds to the currently remaining visual endpoint error. In sum, according to postdictive motor error, learning remains incomplete because the $CD_V$ signal dissociates from the saccade.

### The dissociation between saccade and $CD_V$ produces differences between inward and outward learning

Another central question in studies of saccade learning is why saccade amplitude adapts more to inward target steps than to outward target steps (*Kojima et al., 2004*; *Panouillères et al., 2009*; *Pélisson et al., 2010*). Our experimental data is consistent with this literature as the remaining visual endpoint error $V_2$ is larger after outward learning than after inward learning (*Figure 5F*). We can now examine how the postdiction model explains this difference.

The CD gain $\omega_{cd}$ increases during CTS$_{in}$ learning and decreases during CTS$_{out}$ learning by an equal amount (*Figure 3C*, t-test of $\Delta\omega_{cd}$ between CTS conditions, corrected for direction, $t_{33} = 0.498$, p=0.622). However, learning of the CD gain started from $CD_V$ hypometry in the baseline situation in both conditions (*Figure 5F*). Thus, the visuomotor system converged to a steady state with a relatively strong $CD_V$ hypometry in the CTS$_{out}$ condition ($-(CD_V(281) - M(281)) = 0.98 \pm 1.12°$) and a relatively small $CD_V$ hypermetry in the CTS$_{in}$ condition ($-(CD_V(281) - M(281)) = -0.22 \pm 1.22°$, t-test between CTS$_{in}$ and CTS$_{out}$ corrected for direction $t_{33} = 2.01$, p=0.052). According to postdiction-based learning, this corresponds to a relatively large visual endpoint error in saccade direction after CTS$_{out}$ learning ($1.42 \pm 0.63°$) and a relatively small visual endpoint error against saccade direction after CTS$_{in}$ learning ($-0.61 \pm 0.37°$, t-test between CTS$_{in}$ and CTS$_{out}$ corrected for direction $t_{33} = 4.56$, p<0.001). This characteristic difference is often observed between learning from inward vs. learning from outward target steps and directly follows from the postdiction learning dynamics. A consequence of this is that the internal estimate of the saccade (the $CD_V$ signal in *Figure 3A*) matches the true saccade better in the inward learning conditions and dissociates more strongly in the outward learning conditions. In sum, differences between inward and outward learning may result from the hypometry of $CD_V$ in baseline saccades.

## Discussion

We have presented a model of visuomotor learning that explains many essential properties of saccadic adaptation and visual localization by optimizing visuomotor function to minimize an error signal that takes saccade accuracy and visual accuracy simultaneously into account. A crucial aspect of this model is the plasticity in the CD gain, the gain which the forward dynamics model uses to derive the computed displacement of visual space (denoted as $CD_V$) given the corollary discharge in motor coordinates (denoted as $CD_M$) provided by saccade motor control structures. We hence model visual and motor plasticity via three visuomotor gains: First, a visual gain that maps retinal target input onto a spatial map, second, a motor gain that transforms spatial target distance into a motor

command (inverse model), and third, a CD gain that re-transforms a corollary discharge of the motor command into a computed displacement of visual space, that is, the $CD_V$ signal (forward dynamics model).

Our study addressed the questions which visuomotor representations are affected during saccadic learning, and which error signal can explain this plasticity. We found, firstly, that spatial target perception and $CD_V$ collectively learn from error together with the motor command and, secondly, that this error relies on a postdictive update of space after saccade landing. Accordingly, $CD_V$ accuracy shapes saccade accuracy which provides an emergent explanation for the long-known, yet unexplained observation that saccades typically undershoot their targets and do not fully compensate for peri-saccadic target steps. Third, $CD_V$ was initially hypometric, consistent with saccade hypometry. Fourth, during learning, $CD_V$ dissociated from the saccade, consistent with incomplete motor compensation. Fifth, $CD_V$ underestimated the saccade after learning from outward target steps and overestimated the saccade after learning from inward target steps.

## The corollary discharge pathway contributes to visuomotor learning

Our study provides the first direct evidence on how corollary discharge is integrated via the $CD_V$ signal into space perception in order to guide trans-saccadic motor learning. That $CD_V$ contributes to saccade control and trans-saccadic perception has been consensus since the experimental inactivation of the CD pathway was shown to affect the execution of rapid saccade sequences (*Sommer and Wurtz, 2002*; *Sommer and Wurtz, 2004a*; *Sommer and Wurtz, 2004b*; *Sommer and Wurtz, 2006*; *Wurtz, 2018*). Hence, $CD_V$ information is essential for a predictive update of the spatial map before post-saccadic visual feedback is available. However, motor learning explicitly relies on post-saccadic feedback and occurs in response to systematic feedback change. How $CD_V$ is combined with post-saccadic feedback for error evaluation and whether it correctly reflects saccade changes has remained a question under debate (*Bahcall and Kowler, 1999*; *Collins et al., 2007*; *Collins et al., 2009*; *Zimmermann and Lappe, 2009*; *Schnier et al., 2010*). Supporting *Cavanaugh et al., 2016*, our results suggest that $CD_V$ is available after saccade landing and is integrated with post-saccadic feedback to learn from errors.

## Decoupling of $CD_V$ and saccade in response to artificial target steps

Our results show that $CD_V$ decouples from the saccade during learning from double-step trials, such that it does not match the truly performed saccade (*Bahcall and Kowler, 1999*; *Collins et al., 2009*). Thus, the forward dynamics model that transforms the copy of the motor command ($CD_M$) into the displacement of visual space ($CD_V$) must operate with a non-unity CD gain, suggesting plasticity of the forward dynamics model during learning. One possibility is that the transformation is always performed with a gain that keeps $CD_V$ at baseline level, regardless of adaptation, as proposed by *Bahcall and Kowler, 1999* and *Collins et al., 2009*. However, this would predict that the change in visual localization would strictly follow the change in saccade amplitude. This is not the case in our present data, nor in that of previous studies (*Bahcall and Kowler, 1999*; *Awater et al., 2005*; *Collins et al., 2007*; *Collins et al., 2009*; *Zimmermann and Lappe, 2009*; *Schnier et al., 2010*). Instead, our postdiction model proposes that the CD gain in the forward dynamics model is learned in conjunction with the motor and visual gains with the overall goal to keep the error as small as possible. The decoupling of $CD_V$ and saccade is a consequence of this learning regime.

## $CD_V$ accuracy explains saccade accuracy

Our findings show that $CD_V$ accuracy shapes saccade accuracy in visuomotor steady states. This means that systematic inaccuracies of saccades stem from inaccuracy of internal movement representations. Saccade hypometry (with respect to the target; *Robinson, 1973*; *Henson, 1979*; *Becker, 1989*) is consistent with $CD_V$ hypometry (with respect to the saccade) in natural saccades. That the CD gain in the forward dynamics model is less than one is supported by studies on passive eye rotation (*Bridgeman and Stark, 1991*), visual afterimages (*Grüsser et al., 1987*), and trans-saccadic apparent motion perception (*Szinte and Cavanagh, 2011*; *Zimmermann et al., 2018*). The hypometry of saccades has long been considered a puzzling phenomenon, given that we perform more than 100,000 saccades every day and should have had ample opportunity to learn to get saccades on target. It has been argued that saccades purposefully undershoot their target because

corrective saccades require less effort if executed in the same direction as the primary saccade (*Harris, 1995*). If this were the case, the visuomotor system should actively maintain this undershoot. Yet, learning from inward stepping targets converges at an earlier stage when the visual endpoint error and hence, the corrective saccade, are in opposing direction of the primary saccade (*Kojima et al., 2004*; *Panouillères et al., 2009*). Postdiction-based learning provides a global explanation for visuomotor steady states, both for the hypometry in natural saccades and for the early convergence after learning from experimentally induced target steps.

## Visuomotor learning and spatial stability

Our experimental data were clearly inconsistent with the hypothesis that saccade learning is driven by visual prediction error (*Bahcall and Kowler, 2000*; *Wong and Shelhamer, 2011*; *Collins and Wallman, 2012*). Visual prediction error indicates either the amount by which the world moved, and hence, by which visual stability was violated, or the amount by which the target was unintentionally missed because of some failure in the visuomotor process. The first interpretation fails to explain why peri-saccadic object displacements remain undetected (*Volkmann et al., 1968*; *Bridgeman et al., 1975*; *Li and Matin, 1990*). Indeed, there is cumulative evidence that violations between the post-saccadic percept and the prediction are sacrificed to the assumption of trans-saccadic stability of the world (*Deubel et al., 1996*; *Collins et al., 2009*; *Atsma et al., 2016*; *Jayet Bray et al., 2016*). The second interpretation would predict that the baseline saccade hypometry is restored during learning. However, the visual endpoint error differs between baseline and adapted saccades which is, in turn, well explained by postdiction-based learning.

Postdiction-based learning implies that the visuomotor system assumes trans-saccadic stability as a null hypothesis (see also *Deubel et al., 1996*; *Sommer and Wurtz, 2008*; *Wurtz, 2008*; *Burr and Morrone, 2012*). Hence, errors are always internally attributed, for example to a deficient motor command in the light of changing eye dynamics or to inaccuracies in the visual target localization. Postdiction is a backward modeling process that transforms the post-saccadic visual input into pre-saccadic coordinates using the $CD_V$ signal. Postdiction from post-saccadic input is suitable to update pre-saccadic target location only if the world is assumed stable across the saccade. This assumption makes sense: If the target appears stable in the pre- and the post-saccadic viewing periods, a trans-saccadic target shift is more likely due to internal physiological or neuronal disruptions than to sudden target movement only within the 30–50 ms duration of a saccade. Hence, postdiction-based learning is consistent with saccadic suppression of displacement (SSD; *Volkmann et al., 1968*; *Bridgeman et al., 1975*; *Li and Matin, 1990*) and can keep motor behavior well calibrated as long as the stability assumption is not a fallacy.

## Visuomotor learning and saccadic suppression of displacement

As spatial stability and the question of credit assignment in visuomotor learning are closely linked, it appears interesting to further examine what implications our modeling approach may have for saccadic suppression of displacement (SSD). One of the most prominent findings in the literature is that SSD is abolished, and hence peri-saccadic displacement is correctly detected, if the peri-saccadic target step is blanked for around 250 ms after saccade landing (*Deubel et al., 1996*; *Deubel et al., 2004*; *Collins et al., 2009*; *Srimal and Curtis, 2010*). Thus, information about the target displacement must be available to the visual system in principle but is not used for displacement detection immediately after the saccade. At the same time, saccadic motor learning declines with the introduction of a blanking interval (*Bahcall and Kowler, 2000*; *Fujita et al., 2002*; *Srimal and Curtis, 2010*; *Collins, 2014*). Thus, as speculated above, when the post-saccadic target is used for postdiction after saccade landing, its deviation from the predicted position is sacrificed to a stability assumption for the purpose of fast learning, and lost to perception (*Niemeier et al., 2003*; *Collins et al., 2009*). Since learning from postdictive motor error involves simultaneous updates to motor, visual and $CD_V$ representations, overall consistency of this process may override perception of displacement. Then, the post-saccadic target is taken to be the true target and used to postdictively update the pre-saccadic target position. Blanking would interrupt this process: As the post-saccadic target is not readily available, no learning can occur, perception of the pre-saccadic target position is left as it was, and later comparison to the delayed post-saccadic target becomes possible and displacement visible. It would seem likely, however, that learning may break down if large post-saccadic errors are observed

that are unlikely to be credited to own visuomotor failure (*Wei and Körding, 2009*). Such large errors would also exceed the threshold for displacement detection in SSD experiments.

In our model, $CD_V$ dissociates from the actually performed saccade during learning. This dissociation may constrain SSD in the course of learning. SSD is assumed to be driven by $CD_V$, for example via a pathway from SC superficial layers to the inferior pulvinar, parietal cortex and MT (*Robinson and Petersen, 1985*; *Thiele et al., 2002*; *Wurtz et al., 2011*). If in the course of outward learning, the forward dynamics model underestimates the actually performed saccade along these pathways, the probability to detect the target step may increase. Experiments measuring displacement detection in post-saccadic blanking conditions after saccadic adaptation appear consistent with a learning-based shift of the point of subjective equality for SSD (*Collins et al., 2009*). However, a dissociation between $CD_V$ and saccade may also limit the amount of adaptation. An increasing probability to detect the target step during outward learning may reduce the learning progress.

## Optimizing the saccade vs. optimizing visual predictions

The two learning rules that we compared differ in regard to the error function they aim to minimize during learning. Minimization of visual prediction error aims to optimize visual predictions. This aim is reached when the saccade lands in the expected location, even if that location is different from the saccade target. Hence, it is not robust against distortions within the motor system. For instance, if parts of the inverse model fail because of injury, the saccade may vastly miss the target. However, as long as the forward dynamics model stays accurate and correctly predicts the saccade, there is no need for adjusting the saccade vector since the post-saccadic visual error was correctly predicted. Hence, by its nature the prediction model is not aiming to reach the target. The postdiction model, in contrast, evaluates the error in reaching the original target, estimated by combining $CD_V$ with the post-saccadic error. The postdicted target position correctly indicates the position of the target such that the inverse model adapts to reverse the injury effect.

On a more global level, the visual prediction error leaves open the question what the goal of the saccade is, that is, to which location the gaze should be directed. It only aims to minimize the mismatch between prediction and outcome. Thus, models based on visual prediction error often include an undershoot term, that is, a target location away from the physical target location to account for the observed data. The postdiction model, in contrast, uses internal prediction and post-saccadic error to retroactively evaluate the location that the gaze should have aimed for and sets that as the goal of future saccades. Thus, it predicts where the target should be seen, where the saccade should be aimed and which post-saccadic error should be expected in conjunction with each other.

## Neurophysiological implementation of $CD_V$ plasticity

We argue that the plasticity of $CD_V$ occurs in the transformation of the corollary discharge of the motor command ($CD_M$) by a forward dynamics model from motor to visual coordinates before it can be used by vision (*Crapse and Sommer, 2008a*; *Wurtz, 2018*). This transformation may be implemented at several stages of the CD pathways:in the FEF (*Umeno and Goldberg, 1997*; *Crapse and Sommer, 2008b*; *Crapse and Sommer, 2009*; *Sommer and Wurtz, 2008*; *Zimmermann and Lappe, 2016*), parietal or occipital cortex (*Berman et al., 2017*; *Wurtz et al., 2011*), thalamic nuclei (*Middleton and Strick, 2000*; *Gaymard et al., 2001*; *Zimmermann et al., 2015*), the cerebellum (*Wolpert et al., 1998*; *Chen-Harris et al., 2008*; *Ishikawa et al., 2016*), or in the basal ganglia (*Sommer and Wurtz, 2008*; *Wurtz, 2008*).

The perceptual effects that we found during learning are similar to those obtained from perturbation of the CD pathway from SC via MD thalamus to the FEF. A bias in trans-saccadic target perception occurs when MD thalamus is inactivated (*Cavanaugh et al., 2016*), lesioned (*Ostendorf et al., 2010*), or when the FEF are stimulated via TMS (*Prime et al., 2010*; *Ostendorf et al., 2012*) or subthreshold microsimulation (*White and Snyder, 2007*). The forward dynamics model may be established as the CD information is transmitted via the SC-MD-FEF pathway. Since SC activity stays roughly unaltered during saccadic learning (*Frens and Van Opstal, 1997*; *Edelman and Goldberg, 2002*; *Quessy et al., 2010*), the forward dynamics model is likely implemented at a later stage, for example in the FEF (*Sommer and Wurtz, 2008*; *Crapse and Sommer, 2008a*; *Gerardin et al., 2012*). Beyond that, several cortical areas have been identified to be involved in saccadic learning (*Blurton et al., 2012*; *Guillaume et al., 2018*), which could thus mediate plasticity of the forward

dynamics model. It is also possible that plasticity occurs subcortically in MD thalamus. It was recently shown that MD thalamus is not a passive relay station but actively assembles SC inputs before CD information is transmitted to the FEF (*Cavanaugh et al., 2020*).

Besides the SC-MD-FEF pathway, forward modeling of CD also occurs in the cerebellum (*Wolpert et al., 1998*; *Ishikawa et al., 2016*) to enable feedforward saccade control (*Chen-Harris et al., 2008*; *Ethier et al., 2008a*; *Xu-Wilson et al., 2009*; *Albert and Shadmehr, 2018*). The cerebellum is the main station of oculomotor plasticity and saccadic adaptation (*Kitazawa et al., 1998*; *Barash et al., 1999*; *Golla et al., 2008*; *Xu-Wilson et al., 2009*; *Herzfeld et al., 2015*; *Panouillères et al., 2015*; *Herzfeld et al., 2018*; *Thier and Markanday, 2019*). Hence, a plasticity mechanism for $CD_V$ may also be plausible in the cerebellum. The cerebellum is also a source of a second CD pathway to frontal cortex via the ventrolateral (VL) thalamus (*Middleton and Strick, 2000*). This pathway conveys information from the cerebellum to frontal eye fields and parietal cortex and is involved in saccade learning (*Gaymard et al., 2001*; *Zimmermann et al., 2015*) and spatial updating (*Bellebaum et al., 2005*; *Peterburs et al., 2013*; *Zimmermann et al., 2020*). Forward model computations could also be implemented along that pathway.

## Implications for natural visuomotor behavior

The aim of our study was to broaden the framework of pure motor-based saccadic learning to a framework that also takes changes in perceptual localization into account. In the natural situation, motor errors usually occur due to changes in the eye plant, for example during eye muscle fatigue, ageing or disease. In the laboratory, we simulate the occurrence of motor error by a physical manipulation of the outside world, namely an artificial peri-saccadic target step. We conducted our study in a completely dark laboratory, a visually poor environment, to ensure that any visual references for post-saccadic comparison to the stepped target are eliminated. Our results suggest that the manipulation in the laboratory was successful such that inverse model and forward dynamics model adapt as if the error is inaccurately attributed to own visuomotor failure. In case of an outward target step, the inverse model increases its gain to produce a stronger motor command that compensates for eye muscle fatigue, and the forward dynamics model decreases its gain in order not to overestimate the saccade from the stronger motor command.

However, in the laboratory stetting of our study, the saccade has actually lengthened such that the output of the forward dynamics model, that is, $CD_V$, decouples perception from the actually performed saccade. For the natural situation, when a change in visual feedback is truly due to changing eye dynamics, the plasticity of the forward dynamics model serves to keep $CD_V$ accurate such that it matches the actually performed saccade. The forward dynamics model simulates muscle dynamics to transform motor-efferent copies into visuospatial coordinates. Its adaptability to changing muscle dynamics is highly functional to generate spatially accurate movement representations in natural settings (*Bays and Wolpert, 2007*; *Shadmehr et al., 2010*; *Franklin and Wolpert, 2011*). In accordance with *Shadmehr et al., 2010*, we argue that the forward dynamics model is only useful if it produces accurate predictions, and thus needs to be plastic. Our results reveal that this plasticity exists. Beyond that, our results suggest that the discrepancy between actual and perceived space in the laboratory is a result of the manipulation in physical space and may not occur during natural saccadic learning. In this regard, oculomotor learning may be fundamentally different from manual motor learning, for example in reaching adaptation, where forward dynamics models remain well calibrated despite manipulation of visuospatial feedback in the laboratory (*Shadmehr et al., 2010*; *Michel et al., 2018*). However, manipulation of visuospatial feedback in manual learning paradigms is not coupled to saccades and hence, does not typically remain undetected (*Shadmehr et al., 2010*; *Wolpert et al., 2011*).

The participants of our study learned across saccades that were always directed to the same target, that is, of the same retinal distance and direction. In the natural situation, of course, our gaze rather scans across different locations of the outside world. Hence, on the one hand, it will take longer to adapt in response to fatigue or other perturbations in certain directions than in the laboratory. But on the other hand, perturbations in natural viewing should be rather small, or at least, occur gradually over a longer timescale. However, as learning in natural viewing as well as in the laboratory needs to follow a cost function that is minimized, we think that the plasticity that we found in the visuomotor transformations and the coding scheme of the error are well applicable to natural visuomotor learning. Moreover, it has been shown that learning effects for a specific target location

partially transfer to neighbouring target locations and directions (*Frens and van Opstal, 1994*; *Deubel, 1987*; *Noto et al., 1999*; *Collins et al., 2007*; *Schnier et al., 2010*). In addition, learning has produced particularly strong effects in a global adaptation paradigm in which targets were displaced across saccades in random directions (*Rolfs et al., 2010*). It would be interesting to examine potential mechanisms of the transfer to other target locations and directions in future extensions of our model.

## Implications for patients with schizophrenia

The specific role of CD information for accurate visuomotor calibration may provide new insights into visuomotor abnormalities in patients with schizophrenia. For example, it was proposed that CD impairments could be the reason for the difficulties that patients with schizophrenia present in discriminating between their own actions and those of others (*Feinberg, 1978*; *Frith, 1987*; *Ford et al., 2001*). According to our model, deficits in CD signaling would disturb predictive updating for trans-saccadic target localization as well as postdictive updating for accurate visuomotor learning. Hence, it should cause impairments in baseline saccadic behavior as well as in learning from peri-saccadic target steps. Indeed, studies of patients with schizophrenia have reported deficits in predictive updating (*Rösler et al., 2015*; *Thakkar et al., 2015*; *Thakkar et al., 2017*; *Bansal et al., 2018*), instability in saccade control (*Lencer et al., 2017*), and slowness of saccadic adaptation (*Picard et al., 2009*; *Coesmans et al., 2014*; *Lencer et al., 2017*). However, rate of adaptation (*Coesmans et al., 2014*; *Lencer et al., 2017*) and effects of adaptation on perceptual localization appeared normal in patients (*Lencer et al., 2017*).

## Model predictions

Our modeling approach conveys three central results that carry concrete predictions for future testing. Firstly, our model explains baseline and adapted visuomotor steady states via a unique mechanism, that is, minimization of postdictive motor error. This means that the visuomotor system can flexibly adapt to error via transition to a new steady state. Our model executes this transition without a separate, built-in mechanism of decay to a pre-defined baseline state. Thus, de-adaptation in extinction trials without target step would just be another adaptation. Hence, it would be interesting to test whether the learning rates of usual inward adaptation can successfully predict visuomotor behavior during de-adaptation after outward adaptation, and vice versa.

Secondly, and relatedly, our model suggests that there is a manifold of stable states for any given target step (*Figure 5E*). Hence, different combinations of visual, motor and CD gain exist at which the postdictive motor error is nullified. The combination of learning rates for visual, motor and CD gain determines to which of these possible steady states the visuomotor system converges. This could explain why saccades sometimes do not fully revert to the initial baseline steady state during de-adaptation. For example, in a study by *Gremmler and Lappe, 2019*, subjects first adapted to an inward stepping target (100 trials) and then de-adapted while the target stayed at its pre-saccadic position (again 100 trials). During de-adaptation, motor behavior reverted back but appeared to converge at an earlier state that was different from the initial baseline state. Hence, it seems worthwhile to test whether the de-adapted gains lie within the plane of possible steady states without target step, and to examine further parameters that determine the transition between these steady states.

In addition, minimization of postdictive motor error can further be tested during error-clamp trials in which the error is artificially set to zero. Ongoing adaptation should get bogged down if the postdictive motor error is clamped. In contrast, if the visual error is clamped (usual 'error-clamp trials'), visuomotor behavior should adapt until $CD_V$ and the actual saccade vector are realigned.

Thirdly, it appears promising to search for further behavioral signatures of the decoupling of $CD_V$ from the saccade during learning. For example, in a memory-guided double-step task, if a first saccade is executed to the just adapted target, the landing point of the memory-guided second saccade should be biased because the spatial position of the second target is erroneously updated by the $CD_V$ of the first saccade. Neurophysiologically, it would be interesting to test whether $CD_V$-saccade decoupling is reflected in neurons with spatiotopic receptive fields in response to the adapted target position. Spatiotopically tuned neurons have been found in specific areas of associative cortex, including V6 (*Galletti et al., 1993*) and the ventral intraparietal area (VIP; *Duhamel et al.,*

*1997*). Beyond that, $CD_V$-saccade decoupling could also be reflected in a change of spatial updating before the saccade, like in the FEF, LIP, V2, and V3 (*Walker et al., 1995*; *Nakamura and Colby, 2002*; *Tolias et al., 2001*).

## Model choices and limitations

Our model is based on the minimization of an error function. Consistent with basic principle of error-based learning in the sensorimotor system (*Doya, 1999*; *Wolpert et al., 2011*; *Taylor and Ivry, 2014*), we used the gradient descent approach of delta-rule learning (*Widrow and Hoff, 1960*; *Widrow and Stearns, 1985*) to derive the directions and the dependencies of the trial-by-trial updates of the gain values. It is possible to derive similar learning processes and results using simpler approaches. For example, gains in each step can be simply increased or decreased depending on the direction and size of the error. However, according to the perceptual data the forward dynamics model must learn in opposing direction of the inverse model to minimize the error. This characteristic of learning directions is captured by the error gradients. Hence, our present approach has the advantage of a firm mathematical basis.

Current models of motor adaptation often include two learning processes with different time scales, a slow and a fast process (*Smith et al., 2006*; *Kording et al., 2007*; *Ethier et al., 2008b*; *Cassanello et al., 2019*). While these models produce many essential aspects in motor adaptation they do not consider parallel learning processes that can explain the effects of adaptation on visual perception. Our model is currently designed as a single process model for reasons of simplicity. It is easily conceivable to extend the model to two processes with different timescales. Indeed, multiple simultaneous processes have also been proposed to account for some aspects of adaptation-induced perceptual localization effects (*Awater et al., 2005*; *Collins et al., 2007*; *Hernandez et al., 2008*; *Zimmerman and Lappe, 2010*; *Schnier et al., 2010*; *Gremmler et al., 2014*; *Zimmermann and Lappe, 2016*). Such an extension to a two-process approach might allow a more fine-grained investigation of the interdependencies between motor, visual, and CD changes during learning.

## Post-saccadic use of $CD_V$ and proprioceptive signals

The postdiction model proposes the use of $CD_V$ after the saccade to retroactively estimate (postdict) the location of the saccade target in a pre-saccadic reference frame. $CD_V$, thereby, informs the learning process about the planned saccade vector while the post-saccadic visual error informs the learning process about how much the saccade vector missed the target. A further source of information about the the saccade vector may come from proprioceptive eye position signals. The computation of an estimate of the saccade vector from proprioceptive feedback builds up slowly within 150–300 ms after the saccade (*Fuchs and Kornhuber, 1969*; *Wang et al., 2007*; *Morris et al., 2012*; *Zimmermann et al., 2013*), but it can supplant $CD_V$-based information over time (*Ziesche and Hamker, 2011*; *Ziesche and Hamker, 2014*) and establish a spatiotopic representation of saccade targets (*Zimmermann and Lappe, 2011*; *Burr and Morrone, 2012*; *Zimmermann et al., 2013*). Because of the slow build-up of proprioceptive information, $CD_V$-based postdiction might serve spatial perception within the first hundreds of milliseconds after saccade landing, allowing a fast feedback loop on motor error. As fixation durations in natural viewing barely exceed this time period, a use of $CD_V$ for error computation seems plausible to enable continuous visuomotor learning. However, we might speculate that for longer fixation periods, proprioceptive feedback could update error computation with a more reliable estimate of the performed saccade (*Gremmler and Lappe, 2019*).

## From a predictive to a postdictive update of space across saccades

In the broader scope, predictions are essential to overcome feedback delays in the sensorimotor system, especially in saccades where visual feedback is not available until after movement completion. Thus, prediction of spatial movement outcome is crucial to program rapid subsequent actions, for example pre-planned corrective saccades (*Becker and Fuchs, 1969*) or rapid saccade sequences (*Sommer and Wurtz, 2002*; *Sommer and Wurtz, 2004a*; *Sommer and Wurtz, 2004b*; *Sommer and Wurtz, 2006*; *Zimmermann et al., 2018*). Moreover, spatial prediction enables cerebellar feedforward correction of the motor command during saccade execution – a process that is in

turn rapidly adapted by learning from post-saccadic error evaluation (*Chen-Harris et al., 2008*; *Ethier et al., 2008a*; *Xu-Wilson et al., 2009*; *Albert and Shadmehr, 2018*). What we propose in our model is that saccade motor control may encompass a temporal evolution from a predictive to a postdictive update of space across the saccade. Accordingly, prediction of spatial feedback (for feedforward motor control) would be superseded by postdiction (for motor learning) when actual feedback becomes available after movement completion, provided the world is assumed stable across saccades.

## Acknowledgements

We thank Kerstin Richert for help with data collection and Denis Pelisson and Eckart Zimmermann for helpful comments on the manuscript.

## Additional information

### Funding

| Funder | Grant reference number | Author |
|---|---|---|
| Deutsche Forschungsge-meinschaft | LA952/8-1 | Markus Lappe |

The funders had no role in study design, data collection and interpretation, or the decision to submit the work for publication.

### Author contributions

Jana Masselink, Conceptualization, Resources, Data curation, Software, Formal analysis, Investigation, Visualization, Methodology, Writing - original draft, Project administration; Markus Lappe, Conceptualization, Resources, Supervision, Funding acquisition, Methodology, Project administration, Writing - review and editing

### Author ORCIDs

Jana Masselink (iD) https://orcid.org/0000-0001-5495-8618

### Ethics

Human subjects: All subjects gave written informed consent to participation and publication. The experiment was approved by the ethics committee of the Department of Psychology and Sport Science of the University of Münster (protocol number 2015-21-ML).

### Decision letter and Author response

Decision letter https://doi.org/10.7554/eLife.64278.sa1
Author response https://doi.org/10.7554/eLife.64278.sa2

## Additional files

### Supplementary files

- Transparent reporting form

### Data availability

Data is available at https://doi.org/10.5281/zenodo.4588852.

The following dataset was generated:

| Author(s) | Year | Dataset title | Dataset URL | Database and Identifier |
|---|---|---|---|---|
| Masselink J, Lappe M | 2021 | Visuomotor learning from postdictive motor error | https://doi.org/10.5281/zenodo.4588852 | Zenodo, 10.5281/zenodo.4588852 |

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

# Appendix 1

## Simpler models that cannot explain learning

Our model minimizes postdicted motor error by plastic changes to visual, motor, and CD gain. In this section we illustrate why learning cannot be explained by simpler models that are based on minimization of visual error or that preclude plasticity of one of the three gains.

### Minimization of visual error alone cannot explain learning

Learning occurs in response to visual feedback, that is, to the retinal eccentricity of the post-saccadic target, often referred to as visual or retinal (endpoint) error. Although it is obvious that the visuomotor system must use visual error to asses the outcome of the saccade, there is consensus in the literature that the visual error alone cannot drive learning. Learning from visual error alone is inconsistent with many typical findings in the literature and with our data. First, minimization of visual error aims for steady states with error nullification. However, it was repeatedly shown that the visuomotor system accepts a remaining amount of visual error at steady state. Natural saccades are usually hypometric by means that the target is slightly undershot (*Becker, 1989*; *Henson, 1979*; *Robinson, 1973*). Second, learning does not fully compensate for peri-saccadic target steps (*Straube et al., 1997*; *Wallman and Fuchs, 1998*; *Deubel et al., 1986*; *Noto et al., 1999*). Third, minimization of visual error would predict the visuomotor system to learn until the saccade lands on the post-saccadic target in the CTS paradigm. In line with numerous previous studies (*Schnier et al., 2010*; *Straube et al., 1997*; *Wallman and Fuchs, 1998*; *Deubel et al., 1986*; *Moidell and Bedell, 1988*) our results clearly show that learning converges at an earlier stage. *Appendix 1—figure 1* shows simulations of a model that learns by minimization of visual error. Simulations are shown by lines and data are shown by shaded areas. The simulations illustrate that learning does only converge if the visual error is nullified and hence, the saccade lands on the shifted post-saccadic target (CTS paradigms, columns 1–2).

Fourth, minimization of visual error would predict endless learning in the CVE paradigm where the target is shifted with a constant distance to the post-saccadic gaze location, thus keeping the visual error constant (*Appendix 1—figure 1*, CVE paradigms, columns 3–4). However, even if learning can be maintained for a longer time in the CVE paradigm (as shown also in our experiment), previous studies revealed that learning indeed converges at some stage (*Zimmerman and Lappe, 2010*; *Havermann and Lappe, 2010*; *Robinson et al., 2003*). Fifth, if saccades were to purposely undershoot their target by accepting a certain amount of visual error, this undershoot should be actively maintained during learning. However, learning from inward stepping targets converges with a remaining visual error in opposing direction of the primary saccade (*Kojima et al., 2004*; *Panouillères et al., 2009*).

Lastly, and related to section 1.2 below, minimization of visual error alone does not provide any learning for the CD gain $\omega_{cd}(n)$ because the visual error $V_2(n)$ does not depend on $CD_V(n)$, and, hence, does not depend on $\omega_{cd}(n)$. In our basic model, the visual error is:

$$V_2(n) = P_1 + P_d(n) + \epsilon_m(n) - P_M(n) = P_1(1 - \omega_v(n)\omega_m(n)) + P_d(n) \tag{A1}$$

where $n$ is the trial number, $P_1$ is the physical target eccentricity and $P_d(n)$ is the trans-saccadic target displacement resulting from the imposed target shift and the motor execution noise of the saccade $\epsilon_m(n)$. $P_M(n)$ is the physical saccade vector, $\omega_v(n)$ is the visual gain and $\omega_m(n)$ is the motor gain. Minimization of the visual error according to the delta rule results in the following learning of the visuomotor gains:

$$
\begin{aligned}
\omega(n+1) \quad &= \omega(n) - \alpha \frac{\partial |V_2|^2(n)}{\partial \omega(n)} \\
&= \omega(n) - 2\alpha V_2(n) \frac{\partial V_2(n)}{\partial \omega(n)} \\
&= \omega(n) - 2\alpha V_2(n) \begin{pmatrix} -P_1 \omega_m(n) \\ -P_1 \omega_v(n) \\ 0 \end{pmatrix} \\
&= \begin{pmatrix} \omega_v(n) - 2\alpha_v V_2(n)(-P_1 \omega_m(n)) \\ \omega_m(n) - 2\alpha_m V_2(n)(-P_1 \omega_v(n)) \\ \omega_{cd}(n) \end{pmatrix}
\end{aligned} \tag{A2}
$$

Thus, only the visual gain $\omega_v(n)$ and the motor gain $\omega_m(n)$ learn but the CD gain stays at baseline level throughout learning as $\omega_{cd}(n+1) = \omega_{cd}(n)$ (see **Appendix 1—figure 1**). The gradient descent procedure produces zero learning for the CD gain $\omega_{cd}(n)$ as the error signal that is minimized does not depend on $CD_V(n)$. This is clearly inconsistent with our data. The discrepancy between the small shift of the pre-saccadic target localization and the larger shift in the post-saccadic target localization in our data and many other studies (**Schnier et al., 2010**; **Collins et al., 2007**; **Bahcall and Kowler, 1999**; **Gremmler et al., 2014**; **Hernandez et al., 2008**) is quantified in the basic model as the plasticity of the CD gain. Hence, irrespective of the learning rule, the basic model equations and the data necessitate CD gain plasticity. Since the visual error is independent of $CD_V(n)$, the gradient of post-saccadic visual error learning cannot produce learning of the CD gain. This is, in turn, not in line with the data. Consequently, as can be seen in **Appendix 1—figure 1**, there is a large discrepancy between the visual error learning model and the data with respect to post-saccadic visual localization $\hat{V}_2(n)$.

## Plasticity of all three gains

Our model allows plasticity in three gains, the visual gain (the visual map), the motor gain (the inverse model) and the CD gain (the forward dynamics model). This section explains why plasticity in each of those gains is necessary and why models with plasticity in only one or two of those gains cannot explain the full set of data.

Our study approached the question about which gains are plastic with the most straightforward approach our model can provide — by testing each gain change against zero (see main article, **Figure 3C**). To guide a better intuition why our data argue for plasticity of all three gains $\omega_v(n)$, $\omega_m(n)$ and $\omega_{cd}(n)$, we will further illustrate the contribution of each gain to the learning process.

### Plasticity of the visual gain

Our data reveal small changes in the pre-saccadic target localization that were significant in the CVE conditions. This confirms several previous studies on localization changes during learning (**Schnier et al., 2010**; **Moidell and Bedell, 1988**; **Zimmerman and Lappe, 2010**; **Collins et al., 2007**; **Gremmler et al., 2014**; **Hernandez et al., 2008**). These changes on the visual map are captured by the visual gain $\omega_v(n)$ in our model. Without plasticity of the visual gain, the saccade can adapt, however, the model could not explain the changes in pre-saccadic target localization in the CVE conditions, i.e. a condition in which changes in localization occur without making a saccade. Therefore, although the effects on visual localizations are comparatively small, changes in the visual gain are necessary since no other gain changes can explain these effects. At the same time, we note that the most of the other, stronger learning effects rely mostly on plasticity in the other gains.

### Plasticity of the motor gain

The need for plasticity of the motor gain $\omega_m(n)$ is quite intuitive — it allows the saccade vector to increase or decrease. Without plasticity of the motor gain, the the saccade vector could only increase or decrease if the visual gain increases or decreases. This would predict that the visual gain, and consequently the pre-saccadic target localization, changes as much as the saccade vector. Clearly, this is not the case, as the change in pre-saccadic localization is much smaller than the change in the saccade vector in our data and in most studies in the literature (**Schnier et al., 2010**;

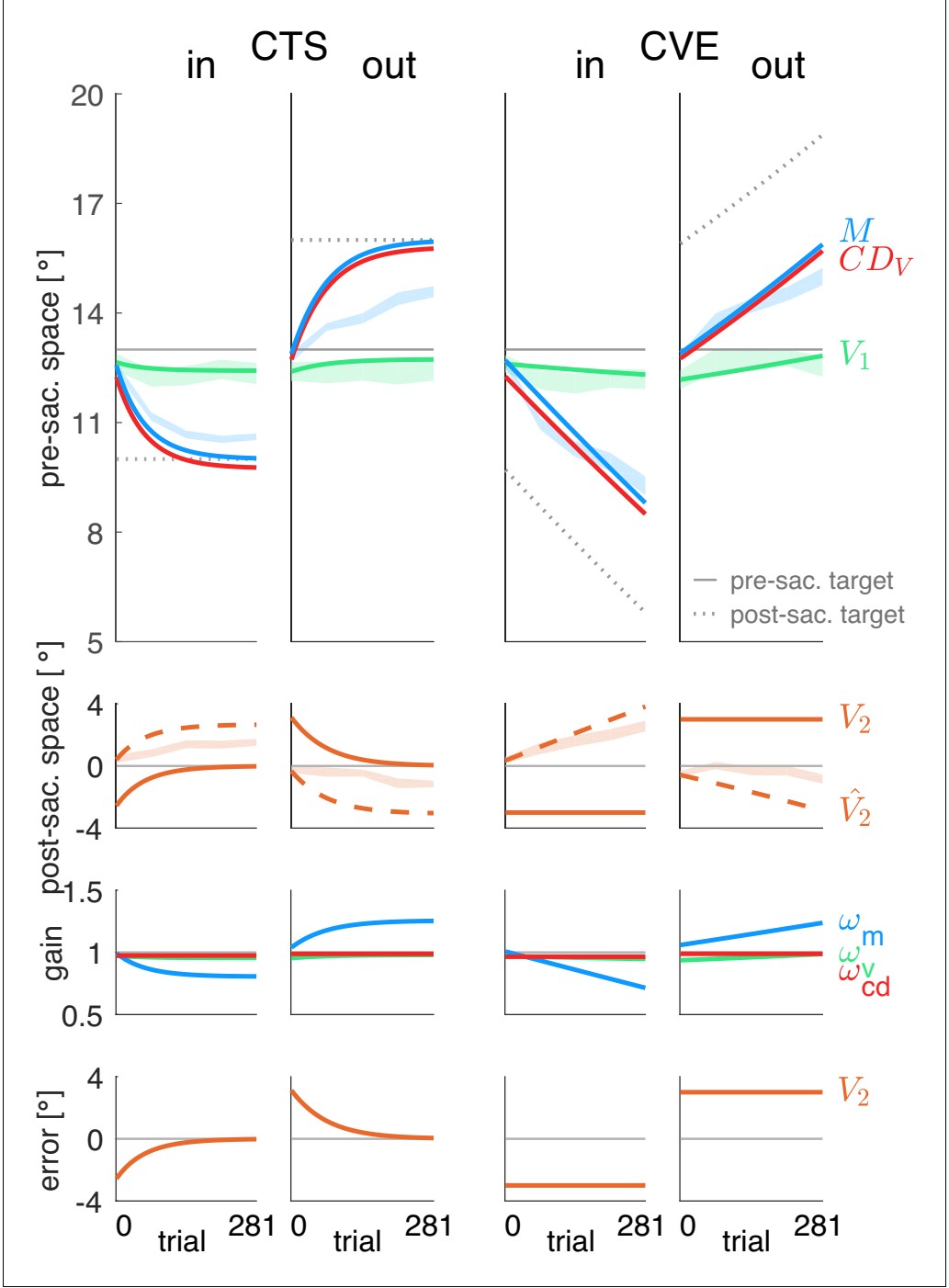

**Appendix 1—figure 1.** Simulations of a model that minimizes visual error. Each column shows the simulation for a specific learning condition. Simulations are shown by lines. Data (means ± standard error) are shown by shaded areas. First row: Visual pre-saccadic target $V_1$ (and pre-saccadic retinal localization data, green shade), motor command $M$ (and saccade vector data, blue shade) and computed displacement of visual space $CD_V$. Second row: Predicted post-saccadic target $\hat{V}_2$ (and post-saccadic retinal localization data, orange shade), visual post-saccadic target $V_2$ (visual error). Third row: Visual gain $\omega_v$, motor gain $\omega_m$, CD gain $\omega_{cd}$. Last row: shows again the visual post-saccadic target $V_2$ (visual error) that is the error signal to be nullified. To nullify $V_2$, the model requires $M$ to learn until the saccade lands on the post-saccadic target. This is not in line with the data. Instead, learning converges at an earlier stage with a remaining, non-zero visual error $V_2$. Hence, the visual error model cannot adequately explain learning. Please note that the CD gain $\omega_{cd}$ stays stable as the visual error $V_2$ does not depend on $CD_V$ such that the gradient descent procedure produces zero learning of the CD gain.

*Zimmerman and Lappe, 2010*; *Collins et al., 2007*; *Gremmler et al., 2014*; *Hernandez et al., 2008*; *Moidell and Bedell, 1988*).

## Plasticity of the CD gain

The plasticity of the CD gain $\omega_{cd}(n)$ (the forward dynamics model) is a main part of our findings and a novel conclusion from our study. The plasticity of the CD gain captures the difference between the pre- and the post-saccadic target localization. This difference reveals a discontinuity in trans-saccadic spatial perception, suggesting that the internally computed displacement of visual space $CD_V(n)$ does not match the actually performed saccade. Without plasticity of the CD gain, the forward dynamics model is fixed and would always compute the same displacement of visual space for the same saccade vector. In that case, the model cannot explain why the post-saccadic target localization dissociates from the pre-saccadic target localization during learning.

*Appendix 1—figure 2* shows simulations of the postdiction model without plasticity of the CD gain such that its learning rate $\alpha_{cd} = 0$. It illustrates that firstly, the model does not capture the discontinuity in trans-saccadic spatial perception because the predicted post-saccadic target position $\hat{V}_2(n)$ does not match the data (shaded area, second row). Secondly, it illustrates that a fixed CD gain $\omega_{cd}(n)$ predicts that the saccade vector adapts until the saccade finally lands on the post-saccadic target. Only in this case is the postdictive motor error $E_{post}(n)$ (first row) nullified. The reason is this: If $\omega_{cd}(n)$ is not plastic and correctly reflects the adapting saccade vector during learning with $\omega_{cd}(n) = 1$, the postdicted pre-saccadic target position $\hat{V}_1(n)$ is equal to the physical post-saccadic target position $P_1 + P_d(n)$ (in pre-saccadic coordinates):

$$
\begin{aligned}
\hat{V}_1(n) &= V_2(n)CD_V(n) \\
&= P_1(1 + \omega_v(n)\omega_m(n)(\omega_{cd}(n) - 1)) + P_d(n) \\
&= P_1(1 + \omega_v(n)\omega_m(n)(1 - 1)) + P_d(n) \\
&= P_1 + P_d(n)
\end{aligned}
\tag{A6}
$$

Thus, in this case, minimizing the postdictive motor error $E_{post}(n)$ equals minimizing the visual post-saccadic error $V_2(n)$ because:

$$
\begin{aligned}
E_{post}(n) &= \hat{V}_1(n) - M(n) \\
&= P_1(1 + \omega_v(n)\omega_m(n)(\omega_{cd}(n) - 2)) + P_d(n) \\
&= P_1(1 + \omega_v(n)\omega_m(n)(1 - 2)) + P_d(n) \\
&= P_1(1 - \omega_v(n)\omega_m(n)) + P_d(n) \\
&= V_2(n)
\end{aligned}
\tag{A7}
$$

This leads to an interesting corollary: Postdictive motor learning without CD gain plasticity is like visual error learning if the internal estimate of the saccade vector $CD_V(n)$ accurately reflects the performed saccade. Postdictive motor learning combines an internal postdiction process via CD with the aim to reach the target.

## Single subject data

*Appendix 1—figures 3–6* present the individual saccade vector, pre- and post-saccadic target localization data of all subjects. Subjects 1–17 performed the inward conditions $CTS_{in}$ and $CVE_{in}$ and subjects 18–35 performed the outward conditions $CTS_{out}$ and $CVE_{out}$.

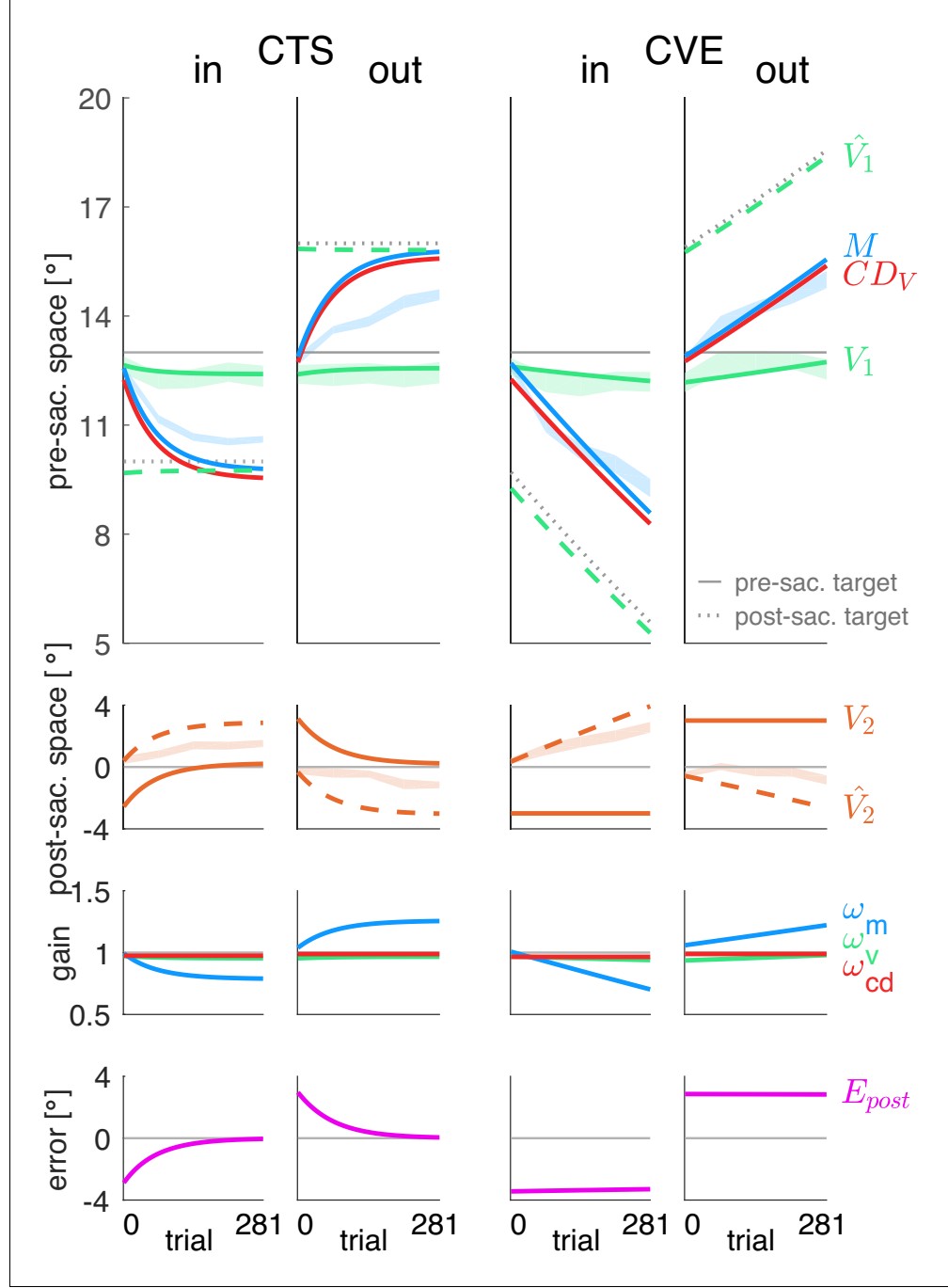

**Appendix 1—figure 2.** Simulations of the postdiction model without plasticity of the CD gain ($\alpha_{cd}$=0). Each column shows the simulation for a specific learning condition. Simulations are shown by lines. Data (means ± standard error) are shown by shaded areas. First row: Visual pre-saccadic target $V_1$ (and pre-saccadic retinal localization data, green shade), motor command $M$ (and saccade vector data, blue shade), computed displacement of visual space $CD_V$, postdicted pre-saccadic target $\hat{V}_1$. Second row: Predicted post-saccadic target $\hat{V}_2$ (and post-saccadic retinal localization data, orange shade), visual post-saccadic target $V_2$. Third row: Visual gain $\omega_v$, motor gain $\omega_m$, CD gain $\omega_{cd}$. Last row: Postdictive motor error $E_{post}$. If the CD gain is not plastic and hence, stays at baseline level as shown here, $CD_V$ almost correctly reflects the saccade vector during learning such that the post-saccadic target localization matches the pre-saccadic target localization. This can be seen in the predicted post-saccadic target position $\hat{V}_2$ that, however, does not match the data (second row). Moreover, a non-plastic CD gain requires the saccade vector to adapt until the saccade lands on the post-saccadic target to nullify the postdictive motor error $E_{post}$. Hence, without CD plasticity the model cannot adequately explain the data.

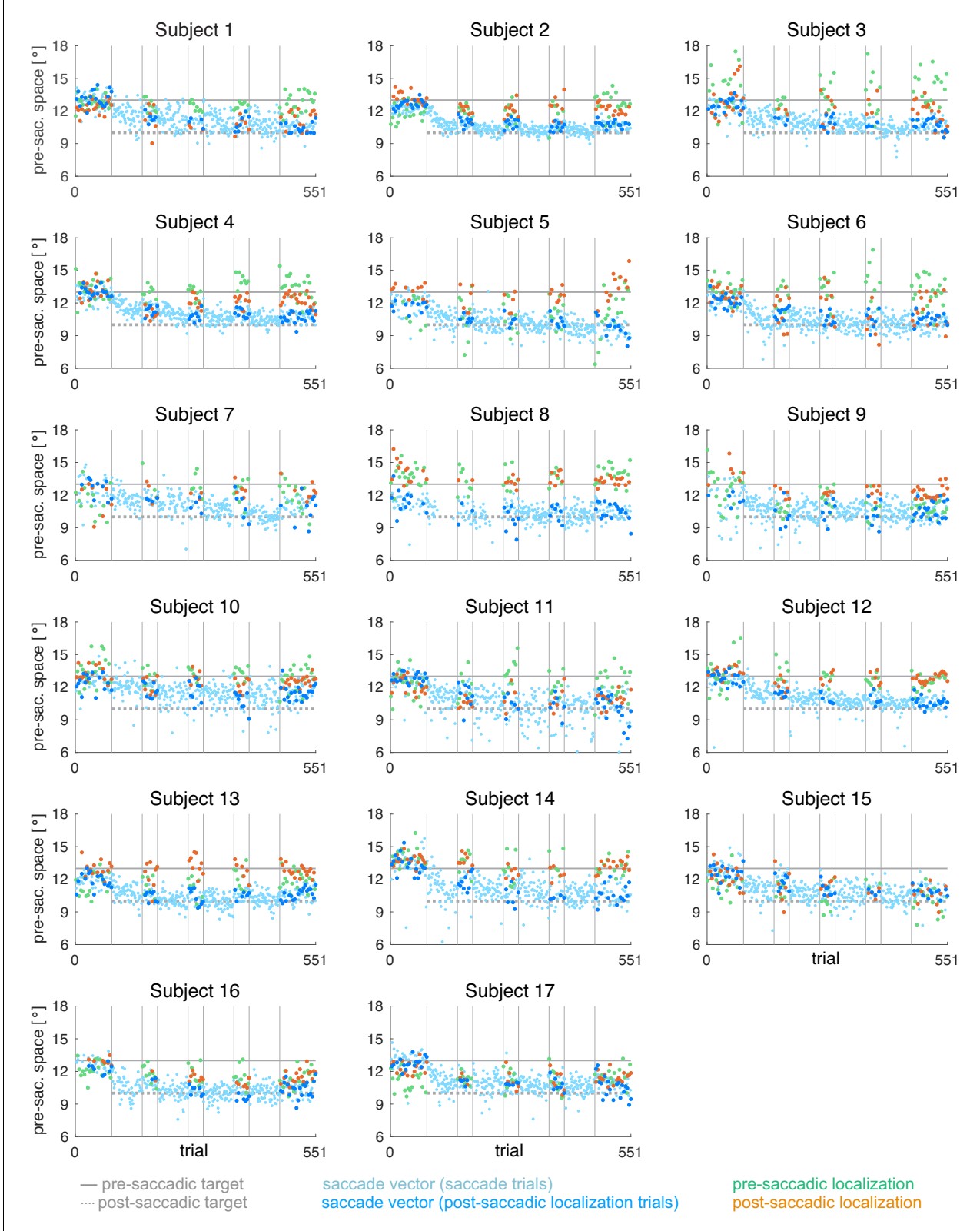

**Appendix 1—figure 3.** Individual subject data for the CTS$_{in}$ condition (subjects 1–17, $N$ = 17). During the saccade, the target was shifted 3° inward (opposite to saccade direction).

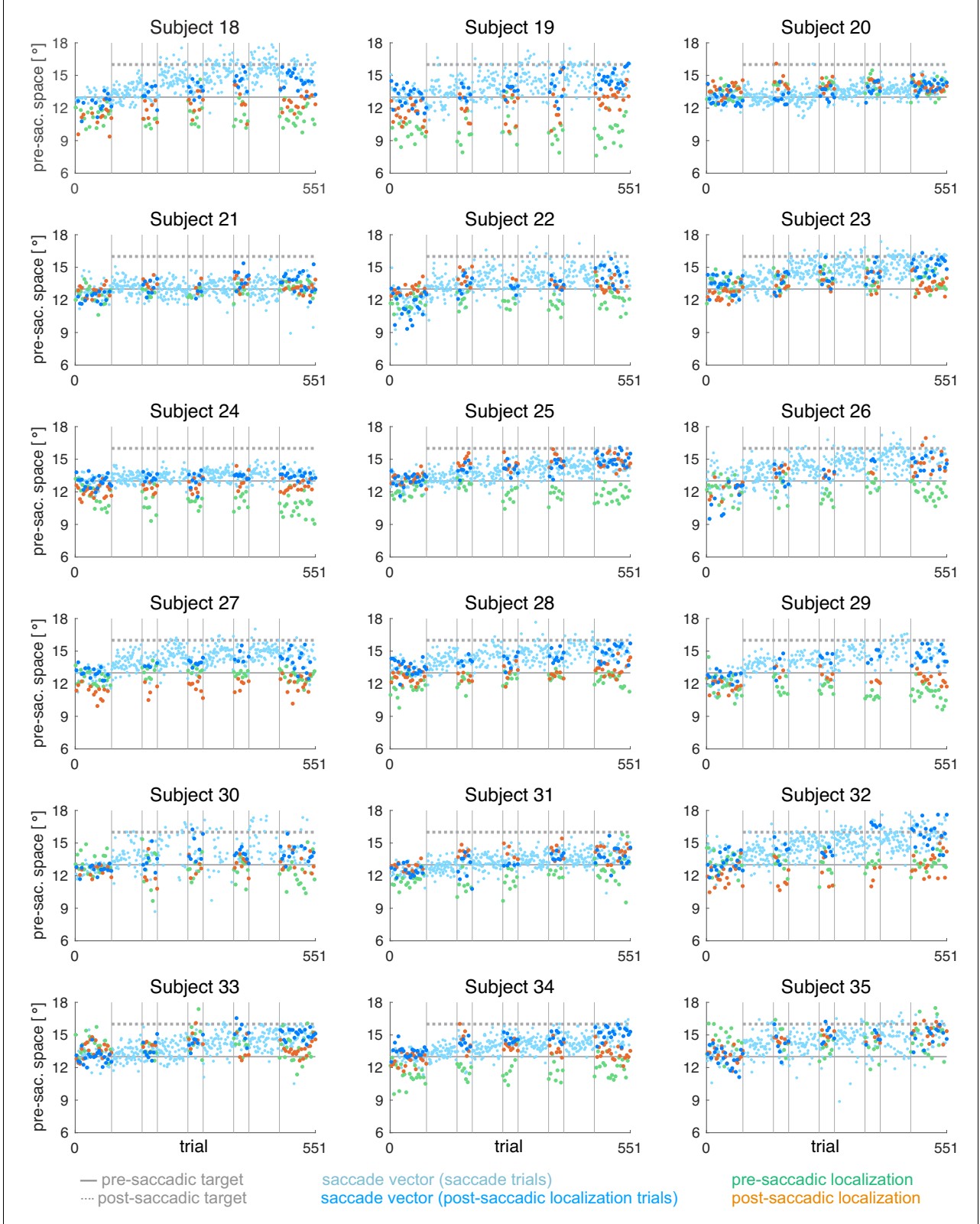

**Appendix 1—figure 4.** Individual subject data for the CTS_out condition (subjects 18–35, *N* = 18). During the saccade, the target was shifted 3° outward (in saccade direction).

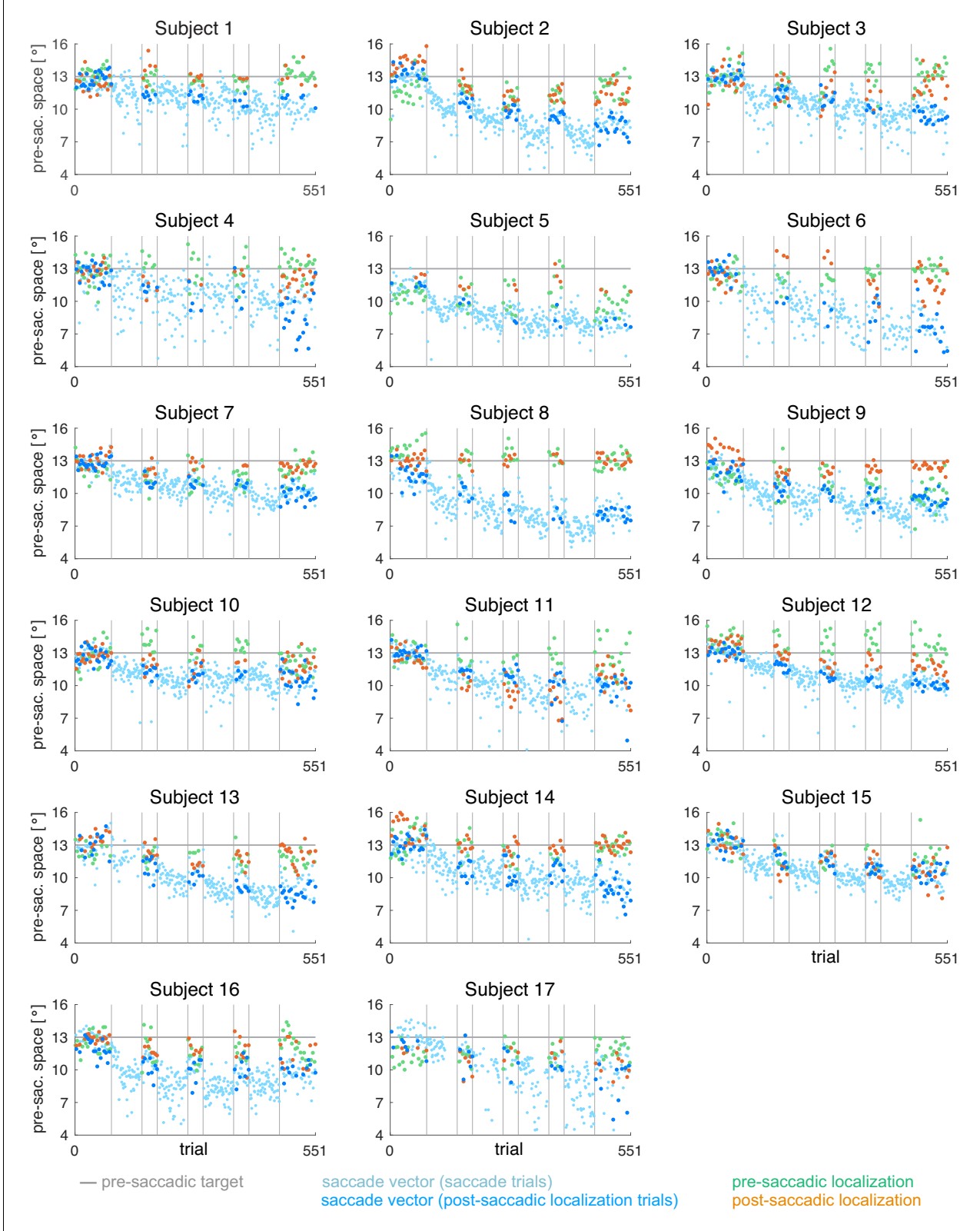

**Appendix 1—figure 5.** Individual subject data for the CVE$_{in}$ condition (subjects 1–17, $N = 17$). During the saccade, the target was shifted to the position that is 3° inward (opposite to saccade direction) of the post-saccadic gaze direction.

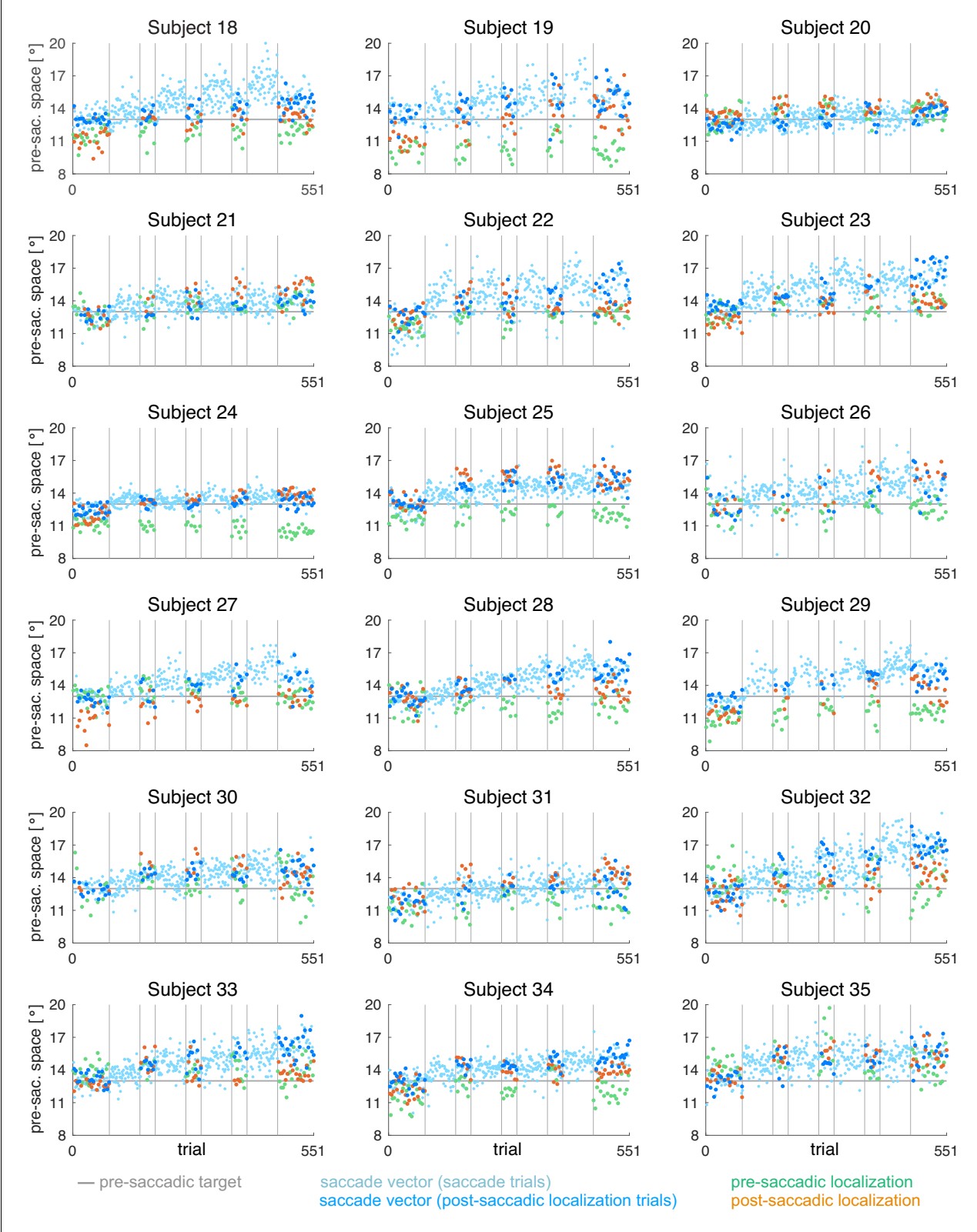

**Appendix 1—figure 6.** Individual subject data for the CVE$_{out}$ condition (subjects 18–35, $N$ = 18). During the saccade, the target was shifted to the position that is 3° outward (in saccade direction) of the post-saccadic gaze direction.

