## [Decision Letter]

**Acceptance summary:**

This paper provides compelling evidence for a new explanation of how post-saccadic visual information could shape sensorimotor learning processes that keep perceptual and motor functions accurate and in sync. It is based on the intriguing idea that learning is driven by updating a pre-saccadic target position by subtracting the corollary discharge signal of the saccade that was just executed from the post-saccadic location of the target. This idea of a postdictive learning signal runs counter to the currently dominant view that saccade adaptation aims to reduce errors in the prediction of post-saccadic target locations. Critically, it explains empirical findings regarding both saccadic adaptation and trans-saccadic localization.

**Decision letter after peer review:**

Thank you for submitting your article "Visuomotor learning from postdictive motor error" for consideration by *eLife*. Your article has been reviewed by three peer reviewers, and the evaluation has been overseen by Miriam Spering as the Reviewing Editor and Tirin Moore as the Senior Editor. The following individuals involved in review of your submission have agreed to reveal their identity: Martin Rolfs (Reviewer #1) and Thérèse Collins (Reviewer #3).

The reviewers have discussed the reviews with one another and the Reviewing Editor has drafted this decision to help you prepare a revised submission.

Summary:

This paper provides evidence for a new explanation of how post-saccadic visual information could shape sensorimotor learning processes that keep perceptual and motor functions accurate and in synch. It is based on the idea that learning is driven by updating a pre-saccadic target position by subtracting the corollary discharge signal of the saccade that was just executed from the post-saccadic location of the target. This idea of a postdictive learning signal runs counter to the currently dominant view that saccade adaptation aims to reduce errors in the prediction of post-saccadic target locations. Critically, it explains empirical findings regarding both saccadic adaptation and trans-saccadic localization. The beauty of the approach is to model three different gains (visual, motor, and corollary discharge gains) in the sensorimotor system at the same time. To achieve this, the authors measure saccade accuracy as well as perceptual localization both before (during fixation) and after saccades at different points in time in the learning process, and in four different adaptation conditions (inward vs outward; constant visual error vs constant target step). In combination, these measures allow the authors to differentiate predictive from postdictive learning regimes as well as their consequences for saccade adaptation and trans-saccadic perception in a single paradigm. The authors then show how their account offer explanations of a range of additional findings repeatedly reported in the literature, based on a simple sensorimotor learning process. These additional phenomena include steady states of adaptation achieved when the error signal in nullified, persistent saccade hypometria, incomplete learning given consistent feedback, and differences between inward- and outward learning.

All reviewers commended the authors on the clarity of the results, and the novelty and importance of their approach. A few substantive concerns were discussed, but these mainly pertain to terminology and some suggestions regarding model assumptions and applications.

Revisions:

1) The most substantive concern shared amongst reviewers regards terminology, specifically, the definition of corollary discharge (CD), a central concept in the report.

First, is CD an estimate of the motor command, or is it a copy of the motor command (Figure 2)? The Wurtz Lab seems to vacillate somewhat on this issue as well so maybe this reflects a confusion in the field (Cavanaugh et al. 2016 say it is a "close copy of the actual movement command"; Cavanaugh et al., 2020 say "…CD is a copy of a neuronal command…"). Related to this, there is also some confusion over what to call the input to the internal model vs. the output of the model. The authors call the output of the internal model CD, so what would the input be called? Lastly, the authors introduce the concept of efference copy which adds to the confusion. The name given to a non-causal copy of the motor command is efference copy or corollary discharge (Crapse and Sommer, 2008, Nature Reviews Neuroscience). CD is then used by the forward model to generate a prediction about the sensory consequences of the movement. Using "CD" as a term for the output of the forward model is unusual and may lead to confusion for readers.

In sum, the reviewers consider this an opportunity to provide clarification of the terminology for the field in the context of the current proposed model.

2) Further to the issue of terminology, the reviewers suggest that the authors call the "internal estimate of the saccade vector" a "visual estimate of saccade vector". Indeed, the estimated saccade vector is a visual representation, i.e. a translation the motor command into an estimate of the distance, in visual space, they eyes have travelled. This makes perfect sense as the output of a forward model, but calling the output a "saccade vector" suggests something motor, and this is confusing.

3) There is also a lack of explicit predictions (tested or suggested) generated by the model, as well as minimal tests of its robustness which are required to show that the results generalize beyond the task. The reviewers think that the manuscript could be more compelling if the authors used the postdictive model and their framework to generate predictions beyond their data set. The reviewers are not asking for follow-up experiments but merely to discuss these ideas for future testing (behaviorally or neurophysiologically). This could be done either as an additional section in the Discussion, or the predictions could be presented as the results are explained in the Results section.

4) How does the postdictive model generalize to natural visual behavior, if at all? The authors touch on this issue, but I am specifically interested in any thoughts on whether the results depend on (arguably unnatural) repeated saccades with the same direction and amplitude in complete darkness.

5) The reviewers would like to see a plot of individual subjects' data. Perhaps use for the format of Figure 3B to separately show the results of two groups of 16 subjects?

6) Finally, the reviewers discussed the possibility of broadening the scope of the paper to another very similar visuo-spatial saccade task in which CD use has been theorized to be crucial: saccadic suppression of displacement (SSD). It seems that extending their model to SSD would be rather easy, and such a generalization would lend credence to it.

The reviewers also suggest that the modeling code is made available, commented with clear references to corresponding equations in the manuscript, and that complete data files of all 32 subjects be made available.

---

## [Author Response]

Revisions:1) The most substantive concern shared amongst reviewers regards terminology, specifically, the definition of corollary discharge (CD), a central concept in the report.First, is CD an estimate of the motor command, or is it a copy of the motor command (Figure 2)? The Wurtz Lab seems to vacillate somewhat on this issue as well so maybe this reflects a confusion in the field (Cavanaugh et al., 2016 say it is a "close copy of the actual movement command"; Cavanaugh et al., 2020 say "…CD is a copy of a neuronal command…"). Related to this, there is also some confusion over what to call the input to the internal model vs. the output of the model. The authors call the output of the internal model CD, so what would the input be called? Lastly, the authors introduce the concept of efference copy which adds to the confusion. The name given to a non-causal copy of the motor command is efference copy or corollary discharge (Crapse and Sommer, 2008, Nature Reviews Neuroscience). CD is then used by the forward model to generate a prediction about the sensory consequences of the movement. Using "CD" as a term for the output of the forward model is unusual and may lead to confusion for readers.In sum, the reviewers consider this an opportunity to provide clarification of the terminology for the field in the context of the current proposed model.

We fully agree that the definition of the corollary discharge signal is a sensitive aspect of our study, and one that suffers from some inconsistencies in the literature. Indeed, navigating these inconsistencies has been part of our difficulty in writing the original manuscript draft. We are very grateful to the reviewers for the encouragement and opportunity to provide clarification to the terminology and we thought hard about how best to do this. In our view, the possible confusion in the field stems from ambiguity concerning the coordinates of CD — whether CD is still in motor coordinates (original definition as a copy of the motor command) or already in visuospatial coordinates (its functional role to update visual representations across saccades). In fact, the step from motor to visuospatial coordinates requires an “explicit recognition that the copy of the motor command, in motor coordinates, must be transformed into sensory coordinates so that it can be compared directly with the sensory input.” (Sommer and Wurtz, 2008). In this regard, the forward modeling process from a pure motor efference copy to the visuospatial movement outcome, i.e. the updated or “predicted” visual representations, actually comprises two steps and not only one. We have refined our terminology as follows (see new Figure 2A).

We start by noting that the corollary discharge is a simple direct copy of the motor command M and that we will denote it as CD_M_. This copy is routed into a forward dynamics model (see Sommer and Wurtz, 2008, Figure 8) that estimates the computed displacement of visual space, denoted as CD_V_, which is also the internal estimate of the saccade vector in visual coordinates. Hence, CD_V_ indicates the expected visual distance that the eyes will travel in response to the copy of the motor command CD_M_, depending on an internal model of the current dynamics of the eyes muscles. In sum, forward modeling comprises the following two steps: First, a forward dynamics model transforms a copy of the motor command (CD_M_) into the computed displacement of visual space (CD_V_). Second, a forward outcome model uses CD_V_ and the visual pre-saccadic target position (V_1_) to predict the visual post-saccadic target position (V^_2_).

We believe that this approach provides a (1) clear description of the process that transforms the copy of the motor command CD_M_ to a visual representation of computed displacement CD_V_, (2) an acknowledgement of the dual aspects of CD that are sometimes mixed-up in the literature, and (3) a simple, consistent and, hopefully, intuitive notation. We have updated Figure 2A according to the new terminology and rearranged the flow in order to better match with typical illustrations in the literature (e.g. Sommer and Wurtz, 2008):

2) Further to the issue of terminology, the reviewers suggest that the authors call the "internal estimate of the saccade vector" a "visual estimate of saccade vector". Indeed, the estimated saccade vector is a visual representation, i.e. a translation the motor command into an estimate of the distance, in visual space, they eyes have travelled. This makes perfect sense as the output of a forward model, but calling the output a "saccade vector" suggests something motor, and this is confusing.

We agree. In line with major point 1 above, we now term the output of the forward dynamics model the computed displacement of visual space, CD_V_. This is equivalent to, or, can be described as a “visual estimate of the saccade vector” like you suggested. We hope that the term “vector” is comprehensible in the sense that it is the internal visuospatial representation of saccade size and does not refer to non-spatial, dynamic motor coordinates.

3) There is also a lack of explicit predictions (tested or suggested) generated by the model, as well as minimal tests of its robustness which are required to show that the results generalize beyond the task. The reviewers think that the manuscript could be more compelling if the authors used the postdictive model and their framework to generate predictions beyond their data set. The reviewers are not asking for follow-up experiments but merely to discuss these ideas for future testing (behaviorally or neurophysiologically). This could be done either as an additional section in the Discussion, or the predictions could be presented as the results are explained in the Results section.

We have added a section on “Model predictions” to our Discussion. Our modeling approach conveys three central results that carry concrete predictions for future testing.

Firstly, our model explains baseline and adapted visuomotor steady states via a unique mechanism, i.e. minimization of postdictive motor error. This means that the visuomotor system can flexibly adapt to error via transition to a new steady state. Our model executes this transition without a separate, built-in mechanism of decay to a pre-defined baseline state. Thus, de-adaptation in extinction trials without target step would just be another adaptation. Hence, it would be interesting to test whether the learning rates of usual inward adaptation can successfully predict visuomotor behavior during de-adaptation after outward adaptation, and vice versa.

Secondly, and relatedly, our model suggests that there is a manifold of stable states for any given target step (Figure 5E). Hence, different combinations of visual, motor and CD gain exist at which the postdictive motor error is nullified. The combination of learning rates for visual, motor and CD gain determines to which of these possible steady states the visuomotor system converges. This could explain why saccades sometimes do not fully revert to the initial baseline steady state during de-adaptation. For example, in a study by Gremmler and Lappe, 2019, subjects first adapted to an inward stepping target (100 trials) and then de-adapted while the target stayed at its pre-saccadic position (again 100 trials). During de-adaptation, motor behavior reverted back but appeared to converge at an earlier state that was different from the initial baseline state. Hence, it seems worthwhile to test whether the de-adapted gains lie within the plane of possible steady states without target step, and to examine further parameters that determine the transition between these steady states.

In addition, minimization of postdictive motor error can further be tested during error-clamp trials in which the error is artificially set to zero. Ongoing adaptation should get bogged down if the postdictive motor error is clamped. In contrast, if the visual error is clamped (usual “error-clamp trials”), visuomotor behavior should adapt until CD_V_ and the actual saccade vector are realigned.

Thirdly, it appears promising to search for further behavioral signatures of the decoupling of CD_V_ from the saccade during learning. For example, in a memory-guided double step task, if a first saccade is executed to the just adapted target, the landing point of the memory-guided second saccade should be biased because the spatial position of the second target is erroneously updated by the CD_V_ of the first saccade. Neurophysiologically, it would be interesting to test whether CD_V_-saccade decoupling is reflected in neurons with spatiotopic receptive fields in response to the adapted target position. Spatiotopically tuned neurons have been found in specific areas of associative cortex, including V6 (Galletti, Battaglini, and Fattori, 1993) and the ventral intraparietal area (VIP; Duhamel, Bremmer, Hamed, and Graf, 1997). Beyond that, CD_V_ -saccade decoupling could also be reflected in a change of spatial updating before the saccade, like in the FEF, LIP, V2 and V3 (Walker, Fitzgibbon, and Goldberg, 1995; Nakamura and Colby, 2002; Tolias et al., 2001).

4) How does the postdictive model generalize to natural visual behavior, if at all? The authors touch on this issue, but I am specifically interested in any thoughts on whether the results depend on (arguably unnatural) repeated saccades with the same direction and amplitude in complete darkness.

We agree that we have to discuss this point in greater detail. On this purpose, we have added the section “Implications for natural visuomotor behavior” to the Discussion.

The aim of our study was to broaden the framework of pure motor-based saccadic learning to a framework that also takes changes in perceptual localization into account. In the natural situation, motor errors usually occur due to changes in the eye plant, e.g. during eye muscle fatigue, ageing or disease. In the laboratory, we simulate the occurrence of motor error by a physical manipulation of the outside world, namely an artificial peri-saccadic target step. We conducted our study in a completely dark laboratory, a visually poor environment, to ensure that any visual references for post-saccadic comparison to the stepped target are eliminated. Our results suggest that the manipulation in the laboratory was successful such that inverse model and forward dynamics model adapt as if the error is inaccurately attributed to own visuomotor failure. In case of an outward target step, the inverse model increases its gain to produce a stronger motor command that compensates for eye muscle fatigue, and the forward dynamics model decreases its gain in order not to overestimate the saccade from the stronger motor command.

However, in the laboratory stetting of our study, the saccade has actually lengthened such that the output of the forward dynamics model, i.e. CD_V_, decouples perception from the actually performed saccade. For the natural situation, when a change in visual feedback is truly due to changing eye dynamics, the plasticity of the forward dynamics model serves to keep CD_V_ accurate such that it matches the actually performed saccade. The forward dynamics model simulates muscle dynamics to transform motor-efferent copies into visuospatial coordinates. Its adaptability to changing muscle dynamics is highly functional to generate spatially accurate movement representations in natural settings (Bays and Wolpert, 2007; Shadmehr et al., 2010; Franklin and Wolpert, 2011). In accordance with Shadmehr et al., 2010, we argue that the forward dynamics model is only useful if it produces accurate predictions and thus, needs to be plastic. Our results reveal that this plasticity exists. Beyond that, our results suggest that the discrepancy between actual and perceived space in the laboratory is a result of the manipulation in physical space and may not occur during natural saccadic learning. In this regard, oculomotor learning may be fundamentally different from manual motor learning, e.g. in reaching adaptation, where forward dynamics models remain well calibrated despite manipulation of visuospatial feedback in the laboratory (Shadmehr et al., 2010; Michel et al., 2018). However, manipulation of visuospatial feedback in manual learning paradigms is not coupled to saccades and hence, does not typically remain undetected (Shadmehr et al., 2010; Wolpert et al., 2011).

The participants of our study learned across saccades that were always directed to the same target, i.e. of the same retinal distance and direction. In the natural situation, of course, our gaze rather scans across different locations of the outside world. Hence, on the one hand, it will take longer to adapt in response to fatigue or other perturbations in certain directions than in the laboratory. But on the other hand, perturbations in natural viewing should be rather small, or at least, occur gradually over a longer timescale. However, as learning in natural viewing as well as in the laboratory needs to follow a cost function that is minimized, we think that the plasticity that we found in the visuomotor transformations and the coding scheme of the error are well applicable to natural visuomotor learning. Moreover, it has been shown that learning effects for a specific target location partially transfer to neighbouring target locations and directions (Frens and Van Opstal, 1994; Deubel, 1987; Noto et al., 1999; Collins et al., 2007; Schnier et al., 2010). In addition, learning has produced particularly strong effects in a global adaptation paradigm in which targets were displaced across saccades in random directions (Rolfs, Knapen, and Cavanagh, 2010). It would be interesting to examine potential mechanisms of the transfer to other target locations and directions in future extensions of our model.

5) The reviewers would like to see a plot of individual subjects' data. Perhaps use for the format of Figure 3B to separately show the results of two groups of 16 subjects?

We have added the plots of the individual subject’s data in the format of Figure 3B to Appendix 1 (Appendix 1—figures 3-6).

6) Finally, the reviewers discussed the possibility of broadening the scope of the paper to another very similar visuo-spatial saccade task in which CD use has been theorized to be crucial: saccadic suppression of displacement (SSD). It seems that extending their model to SSD would be rather easy, and such a generalization would lend credence to it.

We see the reviewers point that a link of the model to SSD would be welcome. We have added the section “Visuomotor learning and saccadic suppression of displacement” to the Discussion:

As spatial stability and the question of credit assignment in visuomotor learning are closely linked, it appears interesting to further examine what implications our modeling approach may have for saccadic suppression of displacement (SSD). One of the most prominent findings in the literature is that SSD is abolished, and hence peri-saccadic displacement is correctly detected, if the peri-saccadic target step is blanked for around 250 ms after saccade landing (Deubel et al., 1996; Deubel, Bridgeman, and Schneider, 2004; Collins et al., 2009; Srimal and Curtis, 2010). Thus, information about the target displacement must be available to the visual system in principle but is not used for displacement detection immediately after the saccade. At the same time, saccadic motor learning declines with the introduction of a blanking interval (Bahcall and Kowler, 2000; Fujita, Amagai, Minakawa, and Aoki, 2002; Srimal and Curtis, 2010; Collins, 2014). Thus, as speculated above, when the post-saccadic target is used for postdiction after saccade landing, its deviation from the predicted position is sacrificed to a stability assumption for the purpose of fast learning, and lost to perception (Niemeier, Crawford, and Tweed, 2003; Collins et al., 2009). Since learning from postdictive motor error involves simultaneous updates to motor, visual and CD_V_ representations, overall consistency of this process may override perception of displacement. Then, the post-saccadic target is taken to be the true target and used to postdictively update the pre-saccadic target position. Blanking would interrupt this process: As the post-saccadic target is not readily available, no learning can occur, perception of the pre-saccadic target position is left as it was, and later comparison to the delayed post-saccadic target becomes possible and displacement visible. It would seem likely, however, that learning may break down if large post-saccadic errors are observed that are unlikely to be credited to own visuomotor failure (Wei and Kording, 2009). Such large errors would also exceed the threshold for displacement detection in SSD experiments.

In our model, CD_V_ dissociates from the actually performed saccade during learning. This dissociation may constrain SSD in the course of learning. SSD is assumed to be driven by CD_V_, e.g. via a pathway from SC superficial layers to the inferior pulvinar, parietal cortex and MT (D. L. Robinson and Petersen, 1985; Thiele, Henning, Kubischik, and Hoffmann, 2002; Wurtz et al., 2011). If in the course of outward learning, the forward dynamics model underestimates the actually performed saccade along these pathways, the probability to detect the target step may increase. Experiments measuring displacement detection in post-saccadic blanking conditions after saccadic adaptation appear consistent with a learning-based shift of the point of subjective equality for SSD (Collins et al., 2009). However, a dissociation between CD_V_ and saccade may also limit the amount of adaptation. An increasing probability to detect the target step during outward learning may reduce the learning progress.

The reviewers also suggest that the modeling code is made available, commented with clear references to corresponding equations in the manuscript, and that complete data files of all 32 subjects be made available.

Yes, we will make the modeling code and subject data available when our manuscript is accepted for publication.